# Primate prefrontal neurons signal economic risk derived from the statistics of recent reward experience

Fabian Grabenhorst[†*], Ken-Ichiro Tsutsui[†‡], Shunsuke Kobayashi[§], Wolfram Schultz

Department of Physiology, Development and Neuroscience, University of Cambridge, Cambridge, United Kingdom

**\*For correspondence:**
fg292@cam.ac.uk

[†]These authors contributed equally to this work

**Present address:** [‡]Division of Systems Neuroscience, Graduate School of Life Sciences, Tohoku University, Sendai, Japan; [§]Department of Neurology, Fukushima Medical University, Fukushima-ken, Japan

**Competing interests:** The authors declare that no competing interests exist.

**Abstract** Risk derives from the variation of rewards and governs economic decisions, yet how the brain calculates risk from the frequency of experienced events, rather than from explicit risk-descriptive cues, remains unclear. Here, we investigated whether neurons in dorsolateral prefrontal cortex process risk derived from reward experience. Monkeys performed in a probabilistic choice task in which the statistical variance of experienced rewards evolved continually. During these choices, prefrontal neurons signaled the reward-variance associated with specific objects ('object risk') or actions ('action risk'). Crucially, risk was not derived from explicit, risk-descriptive cues but calculated internally from the variance of recently experienced rewards. Support-vector-machine decoding demonstrated accurate neuronal risk discrimination. Within trials, neuronal signals transitioned from experienced reward to risk (risk updating) and from risk to upcoming choice (choice computation). Thus, prefrontal neurons encode the statistical variance of recently experienced rewards, complying with formal decision variables of object risk and action risk.
DOI: https://doi.org/10.7554/eLife.44838.001

## Introduction

Rewards vary intrinsically. The variation can be characterized by a probability distribution over reward magnitudes. Economists distinguish between risk when probabilities are known and ambiguity when probabilities are only incompletely known. The variability of risky rewards can be quantified by the higher statistical 'moments' of probability distributions, such as variance, skewness or kurtosis (*Figure 1A*). The most frequently considered measure of economic risk is variance (*D'Acremont and Bossaerts, 2008*; *Kreps, 1990*; *Markowitz, 1952*; *Rothschild and Stiglitz, 1970*), although skewness and even kurtosis constitute also feasible risk measures that capture important components of variability (*Burke and Tobler, 2011*; *D'Acremont and Bossaerts, 2008*; *Genest et al., 2016*; *Symmonds et al., 2010*). Thus, among the different definitions of economic risk, variance constitutes the most basic form, and this study will consider only variance as economic risk.

Existing neurophysiological studies on risk have used explicit, well-established informative cues indicating specific levels of risk in an unequivocal manner (*Fiorillo et al., 2003*; *Lak et al., 2014*; *Ledbetter et al., 2016*; *McCoy and Platt, 2005*; *Monosov, 2017*; *Monosov and Hikosaka, 2013*; *O'Neill and Schultz, 2010*; *O'Neill and Schultz, 2013*; *Raghuraman and Padoa-Schioppa, 2014*; *Stauffer et al., 2014*; *White and Monosov, 2016*). However, in daily life, outside of testing laboratories, animals are confronted with risky rewards without being over-trained on explicit, risk-descriptive cues; they need to estimate themselves the risk from the experienced rewards in order to make economic decisions. Thus, the more natural way to address the inherently risky nature of rewards is to sample the occurrence of reward from experience in a continuous manner, integrate it over time, and compute risk estimates from that information.

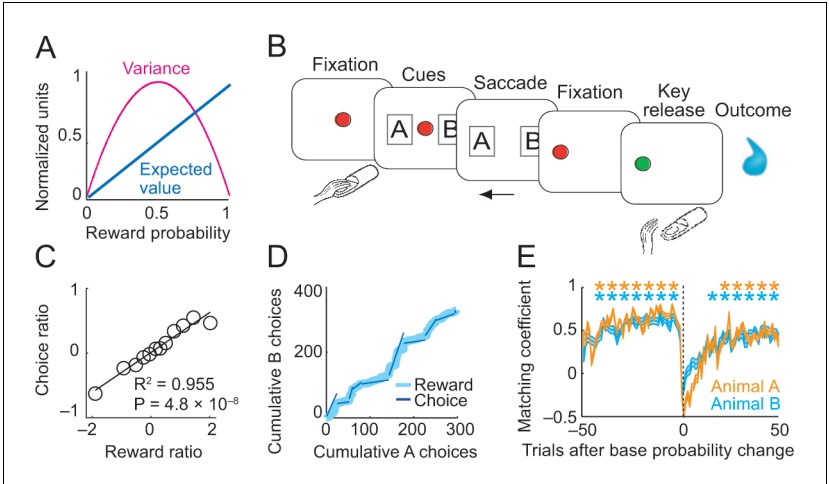

**Figure 1.** Risk, choice task and basic behavior. (**A**) Relationship between risk measured as reward variance and reward probability. (**B**) Choice task. The animal made a saccade-choice between two visual stimuli (fractals, 'objects') associated with specific base reward probabilities. Object reward probabilities varied predictably trial-by-trial according to a typical schedule for eliciting matching behavior and unpredictably block-wise due to base-probability changes. Probabilities were uncued, requiring animals to derive reward risk internally from the variance of recently experienced rewards. Left-right object positions varied pseudorandomly. (**C**) Matching behavior shown in log ratios of rewards and choices. Relationship between log-transformed choice and reward ratio averaged across sessions and animals (N = 16,346 trials; linear regression; equally populated bins of reward ratios; standard errors of the mean (s.e.m.) were smaller than symbols). (**D**) Cumulative object choices in an example session. The choice ratio in each trial block (given by the slope of the dark blue line) matched the corresponding reward ratio (light blue). (**E**) Adaptation to block-wise reward-probability changes. Matching coefficient (correlation between choice and reward ratio) calculated using seven-trial sliding window around base probability changes (data across sessions, asterisks indicate significant correlation, p<0.05).

DOI: https://doi.org/10.7554/eLife.44838.002

The following source data is available for figure 1:

**Source data 1.**
DOI: https://doi.org/10.7554/eLife.44838.003

The current study aimed to obtain a more representative view on neuronal risk processing by studying variance risk estimated from experience. To this end, we examined behavioral and neuro-physiological data acquired in a probabilistic choice task (*Herrnstein, 1961*; *Tsutsui et al., 2016*) in which reward probabilities, and thus risk, changed continuously depending on the animal's behavior, without being explicitly indicated by specific risk-descriptive cues. Similar to previous risk studies, experienced rewards following choices for specific objects or actions constituted external cues for risk estimation. The terms of risk seeking and risk avoidance indicate that risk processing is subjective. Therefore, we estimated subjective risk from the recent history of the animal's own choices and rewards, based on logistic regression; we investigated the objective risk derived from experimentally programmed reward probabilities only for benchmark tests, again without explicit risk-descriptive cues. The individually distinct risk attitudes reflect the fact that risk influences economic decisions (*Holt and Laury, 2002*; *Stephens and Krebs, 1986*; *Weber and Milliman, 1997*). Therefore, we followed concepts of decision-making based on competition between object values or action values (*Deco et al., 2013*; *Sutton and Barto, 1998*; *Wang, 2008*) and defined in analogy object risk as the risk attached to individual choice objects and action risk attached to individual actions for obtaining the objects. We studied individual neurons in the dorsolateral prefrontal cortex (DLPFC) of rhesus monkeys where neurons are engaged in reward decisions (*Barraclough et al., 2004*; *Donahue and Lee, 2015*; *Kennerley et al., 2009*; *Seo et al., 2012*; *Tsutsui et al., 2016*; *Watanabe, 1996*). The results suggest that DLPFC neurons signal risk derived from internal estimates of ongoing reward experiences.

## Results

### Choice task and behavior

Two monkeys performed in a probabilistic choice task (*Figure 1B*) in which they repeatedly chose between two visual objects (A and B) to obtain liquid rewards (*Tsutsui et al., 2016*). The matching behavior in this task has previously been reported (*Tsutsui et al., 2016*). Here, we briefly show the basic pattern of matching behavior across animals before addressing the novel question of how risk influenced behavior and neuronal activity. In the task, both options had the same, constant reward amount but specific, independently set base probabilities during blocks of typically 50–150 trials. Despite the set base probability, each object's instantaneous reward probability increased in each trial in which the object was not chosen but fell back to its base probability after the object had been chosen (*Equation 1*). Importantly, once reward probability had reached p=1.0, the reward remained available until the animal chose the object. Thus, reward probabilities changed depending on the animal's choice, and the current, instantaneous level of reward probability was not explicitly cued. Accordingly, to maintain an estimate of reward probability, the animal would need to track internally its own choices and experienced rewards.

Under these conditions, an efficient strategy consists of repeatedly choosing the object with the higher base probability and choosing the alternative only when its instantaneous reward probability has exceeded the base probability of the currently sampled object (*Corrado et al., 2005*; *Houston and McNamara, 1981*). Aggregate behavior in such tasks usually conforms to the matching law (*Herrnstein, 1961*), which states that the ratio of choices to two alternatives matches the ratio of the number of rewards received from each alternative. Such behavior has been observed in monkeys (*Corrado et al., 2005*; *Lau and Glimcher, 2005*; *Lau and Glimcher, 2008*; *Sugrue et al., 2004*). Consistent with these previous studies, the animals allocated their choices proportionally to the relative object-reward probabilities (*Figure 1C,D*). Through their alternating choices, they detected uncued base-probability changes and adjusted their behavior accordingly (*Figure 1E*).

Thus, the animals' behavior in the choice task corresponded well to the theoretical assumptions (*Herrnstein, 1961*; *Houston and McNamara, 1981*) and suggested that they estimated well the current reward probabilities of the choice options. On the basis of these choices, we derived specific risk measures that we used as regressors for neuronal responses in DLPFC.

### Definition of risk measures

We used two measures of variance risk. The first, objective risk measure linked our study to previous work and provided a foundation for investigating subjective risk. In the choice task, the objective reward probability evolved continually from the animal's choices (*Equation 1*). This characteristic allowed us to calculate objective risk in each trial as statistical variance derived only from the objective reward probability (*Equation 2*) (reward amount was constant and identical for each option). The second, subjective measure of risk addressed the origin of the neuronal risk information in the absence of informative risk cues. Here, risk was derived directly and subjectively from the evolving statistical variance of recently experienced reward outcomes resulting from specific choices (*Equation 5-8*). We then investigated whether responses in DLPFC neurons associated these two different risk estimates with choice objects ('object risk') or with actions required for choosing these objects ('action risk'), irrespective of the animal's choice.

### Neuronal coding of objective risk

The first, most basic risk measure derived risk as variance in each trial and for each choice object directly from the 'inverted-U' function (*Figure 1A*) of the true, objective, 'physical' reward probability (which depended in each trial on the animal's previous choices) (*Equation 2*). This risk measure derived from the programmed binary probability (Bernoulli) distribution on each trial that governed actual reward delivery and assumed no temporal decay in subjective perception, attention or memory of reward occurrence and risk. Risk as variance defined in this way increased between p=0.0 and p=0.5 and decreased thereafter, following an inverted-U function (*Figure 1A*). Thus, probability and risk were not one and the same measure; they correlated positively for the lower half of the probability range (p=0.0 to p=0.5) and inversely for the upper half of the probability range (p=0.5 to p=1.0); higher probability did not necessarily mean higher risk.

Among 205 task-related DLPFC neurons, 102 neurons had activity related to the objective form of risk, defined as variance derived from the true probability (50% of task-related neurons; p<0.05 for object risk coefficient, multiple linear regression, *Equation 3*; 185 of 1222 significant task-related responses in different trial periods, 15%; task-relatedness assessed by Wilcoxon test with p<0.005, corrected for multiple comparisons). During the fixation periods, the activity of the neuron shown in *Figure 2A* reflected the risk for object B (*Figure 2A,B*; p=0.0172, multiple linear regression, *Equation 3*), whereas the coefficient for object-A risk was non-significant (p=0.281). Critically, the signal reflected the variance risk associated with object B and neither reward probability for object A or B (both p>0.36) nor choice, action or left-right cue position (all p>0.25). To further evaluate the object-specificity of the risk response, we classified the response using the angle of regression

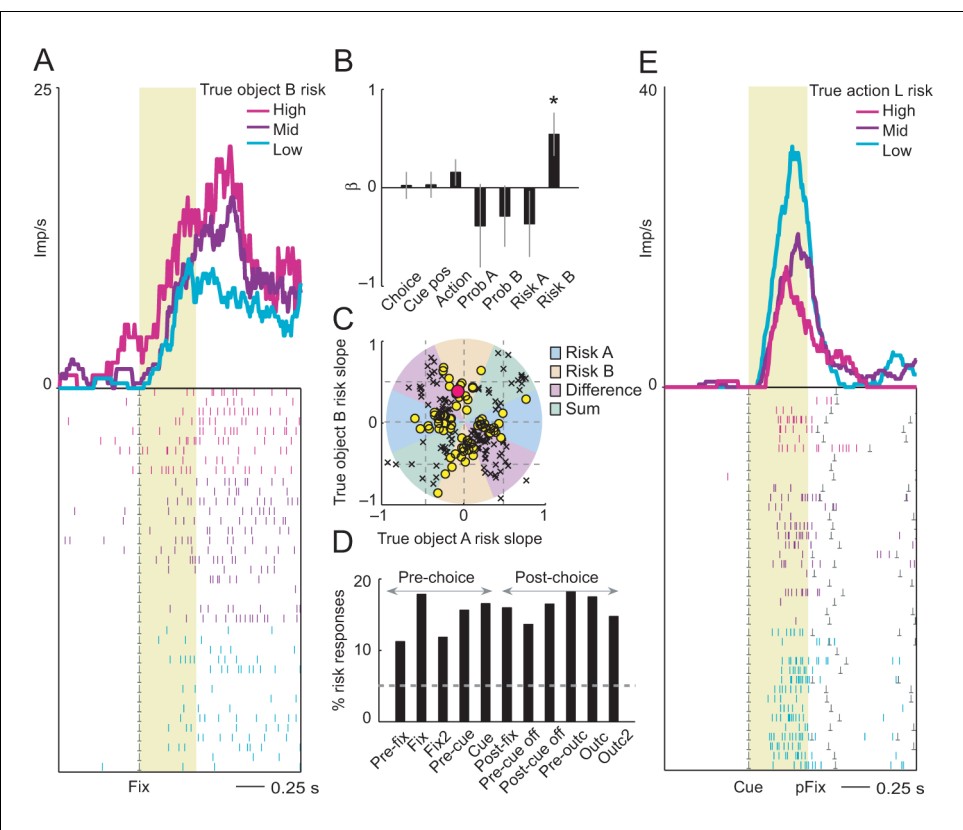

**Figure 2.** Neuronal coding of objective risk for objects and actions. (**A**) Activity from a single DLPFC neuron in fixation period related to the true risk associated with object B, derived from reward probability. Top: peri-event time histogram of impulse rate, aligned to fixation spot onset, sorted into object-risk terciles. Bottom: raster display: ticks indicate impulses, rows indicate trials; gray dots indicate event markers. Yellow shaded zone (500 ms after fixation cue onset) indicates analysis period. (**B**) Beta coefficients (standardized slopes ± s.e.m) from multiple linear regression. Only the object-B risk beta coefficient was significant (p=0.0172, t-test; all other coefficients: p>0.25). (**C**) Categorization of coding risk for object A or B or relative risk (risk difference or risk sum) based on the angle of regression coefficients across neurons. Each symbol represents a neuronal response by its normalized risk slopes for objects A and B; different symbols indicate differently classified neuronal responses as follows. Yellow circles: responses classified as coding object risk; red circle: position of object-B risk response of the neuron shown in A and B; black crosses: responses classified as coding relative risk. (**D**) Percentages of object risk responses for all task epochs (multiple linear regression; 1222 task-related responses from 205 neurons). (**E**) Activity of a single DLPFC neuron related to the true risk for leftward saccades (left action risk) in cue period (pFix: onset of peripheral fixation spot confirming choice).

DOI: https://doi.org/10.7554/eLife.44838.004

The following source data is available for figure 2:

**Source data 1.**

DOI: https://doi.org/10.7554/eLife.44838.005

coefficients (*Equation 4*, see Materials and methods), which confirmed object risk coding rather than risk difference or risk sum coding of the two objects (*Figure 2C*). Thus, the neuron's activity signaled the continually evolving true risk for a specific choice object. Across neuronal responses, classification based on the angle of regression coefficients (*Equation 4*) showed 195 significant risk responses in 89 neurons (p<0.05, F-test), of which 75 responses (39%, 47 neurons, *Figure 2C*) coded object risk rather than risk difference or risk sum, thus confirming the object-risk coding described above. In the population of DLPFC neurons, object-risk responses occurred in all task periods (*Figure 2D*), including the pre-choice periods, in time to inform decision-making.

We also tested for neuronal coding of objective action risk. Once the left-right position of the risky choice objects was revealed on each trial, the animals could assess the risk associated with leftward or rightward saccade actions. A total of 57 DLPFC neurons coded action risk in cue or post-cue task periods (p<0.05; multiple linear regression, *Equation 4*, action risk regressors). During the choice cue phase, the activity of the neuron in *Figure 2E* reflected the action risk for leftward saccades (p=0.041, multiple linear regression, *Equation 4*), whereas the coefficient for rightward saccade risk was non-significant (p=0.78). The signal reflected the variance risk associated with left actions but neither reflected reward probability for left or right actions (both p>0.44), nor the actual choice or left-right action (both p>0.34) but reflected additionally the left-right cue position (p=0.0277).

Taken together, a significant number of DLPFC neurons showed activity related to the true, objective risk associated with specific objects or actions. Although this basic risk measure accounted for the continually evolving risk levels in the choice task, it did not reflect the assumption that the animals' risk estimates were likely subjective, owing to imperfect knowledge and memory of the true reward probabilities. Accordingly, we next defined a subjective risk measure, validated its relevance to the animals' behavior, and tested its coding by DLPFC neurons.

## Subjective risk: definition and behavior

Whereas our first, objective risk measure concerned the variance derived from objective (true, programmed) reward probability, our second risk measure assumed imperfect, temporally degrading assessment of recently experienced rewards. To obtain this measure, we established subjective weights for recently experienced rewards using logistic regression on the animal's reward and choice history (*Equation 5*), following standard procedures for analyzing subjective decision variables in similar choice tasks (*Corrado et al., 2005*; *Lau and Glimcher, 2005*). These weights (*Figure 3A,B*) revealed declining influences of past rewards and past choices on the animal's current-trial choice, in line with previous matching studies (*Corrado et al., 2005*; *Lau and Glimcher, 2005*). On the basis of this result, the assessment of subjective variance risk in each trial considered only data from the preceding 10 trials.

For behavioral and neuronal comparisons between subjective risk and value, we first defined 'object value' as the recency-weighted reward value for a specific choice object (*Tsutsui et al., 2016*). We followed previous studies of matching behavior (*Lau and Glimcher, 2005*) that distinguished two influences on value: the history of recent rewards and the history of recent choices. The first value component related to reward history can be estimated by the mean of subjectively weighted reward history over the past ten trials (*Figure 3C*, dashed blue curve, *Equation 6*) and provided a useful comparison for our subjective risk measure, which also derived from reward history (described in the next paragraph). To estimate a comprehensive measure of object value for behavioral and neuronal analysis, we incorporated the additional effect of choice history on value, which is distinct from reward history as shown in previous studies of matching behavior (*Lau and Glimcher, 2005*). Thus, we estimated object value based on both subjectively weighted reward history and subjectively weighted choice history (*Equation 7*); this constituted our main value measure for behavioral and neuronal analyses. (We consider distinctions between reward and choice history and their potential influence on risk in the Discussion).

We next calculated for each trial the subjective measure of 'object risk' as the statistical variance of the distribution of rewards of the preceding ten trials, separately for each object (*Figure 3C*, dashed magenta curve; *Equation 8*). Specifically, subjective object risk was derived from the sum of the weighted, squared deviations of object-specific rewards from the mean of the object-specific reward distribution over the past ten trials (*Equation 8*); subjective weighting of the squared deviations with empirically estimated reward weights (*Figure 3A*, *Equation 5*) accounted for declining

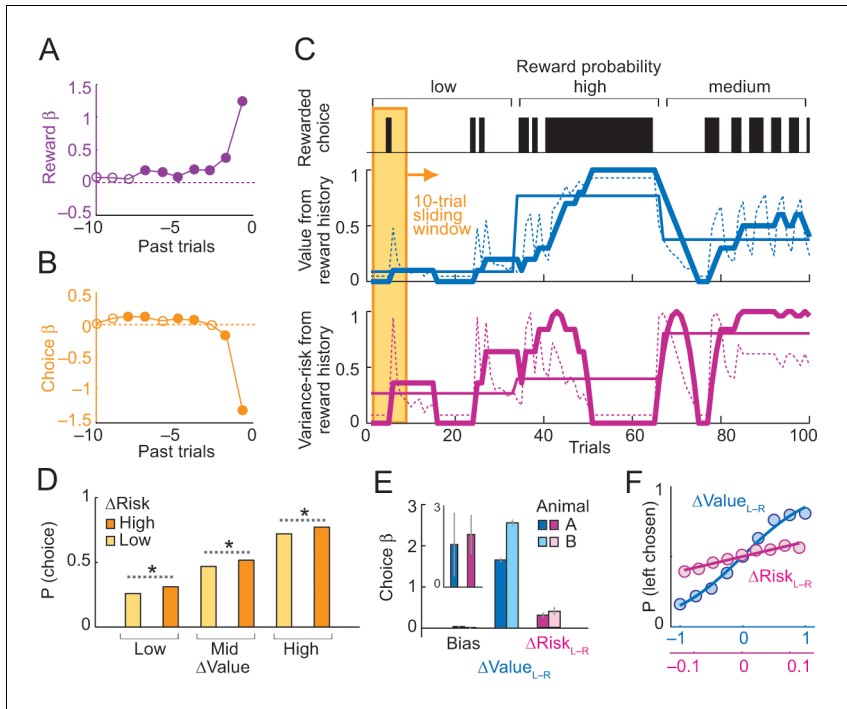

**Figure 3.** Deriving subjective risk from the variance of reward history. (**A**) Subjective weights of influence of recent rewards on object choice, as derived from logistic regression. Filled symbols indicate significance (p<0.005, t-test; pooled across animals). (**B**) Subjective weights of influence of recent choice on object choice, as derived from logistic regression. (**C**) Approach for deriving subjective risk from the variance of recent reward history. Upper panel: vertical black bars represent rewarded choices for object A. Middle/lower panels: trial-by-trial estimates of value (middle) and risk (lower) calculated, respectively, as mean and variance of reward history over last 10 trials (using weights shown in **A**). The dashed magenta line indicates the subjective risk estimate used for neuronal and behavioral analysis. Heavy lines: running average of weighted estimates. Thin solid lines: unweighted, objective value/risk. Value from reward history was highest in the high-probability block, whereas risk was highest in medium-probability block (inverted U-shaped relationship, see *Figure 1A*). All units were normalized to allow for visual comparisons. (**D**) Positive influence of subjective risk on choice (risk-seeking attitude) and separate value and risk influences on choice. Object-choice probability increased with risk difference between objects (ΔRisk, sorted by median split; 'high' indicates higher risk with object A compared to object B; p<0.002 for all pair-wise choice probability comparisons between adjacent relative-risk levels, $\chi^2$-tests; N = 16,346 trials). The risk effect added monotonically and consistently to the increase of choice probability with object value difference (ΔValue). (**E**) Logistic regression. Coefficients (β) for relative value (ΔValue, p=4.4 × $10^{-39}$), relative risk (ΔRisk. p=2.8 × $10^{-4}$) and left-right bias (Bias, p=0.698) across sessions (t-tests, random-effects analysis). The constant (bias) was not significant, suggesting negligible side bias. The inset shows coefficients for a subset of trials where value difference was minimized (10% of trials); only risk difference was significantly related to choice (p=0.0072) but not value difference (p=0.182), data pooled across animals. (**F**) Psychometric functions relating relative value and risk to choice probability (across animals and sessions).

DOI: https://doi.org/10.7554/eLife.44838.006

The following source data and figure supplements are available for figure 3:

**Source data 1.**
DOI: https://doi.org/10.7554/eLife.44838.009
**Figure supplement 1.** Influences on saccadic reaction times.
DOI: https://doi.org/10.7554/eLife.44838.007
**Figure supplement 1—source data 1.**
DOI: https://doi.org/10.7554/eLife.44838.008

influences of more remote past trials on behavior. Object risk defined in this manner varied continuously as a function of past rewards and past choices as follows. When block-wise reward probability was low, each reward increased both object risk and value; without further rewards, both risk and value decreased gradually over subsequent trials (*Figure 3C* compare blue and magenta curves;

note that the blue curve shows the effect of reward history on value). When reward probability was high, object risk increased when a reward was omitted (which drove the instantaneous probability toward the center of the inverted-U function); with further rewards, object risk decreased gradually over subsequent trials (driving the instantaneous probability toward the high end of the inverted U function). Risk was highest in medium-probability blocks with alternating rewarded and unrewarded trials (variations around the peak of the inverted-U function).

We next assessed the animals' risk attitude by testing the influence of subjective risk on the animals' choices. Choice probability increased monotonically with increasing difference in object value (*Figure 3D*, ΔValue, derived from the sum of weighted reward and choice histories, *Equation 7*). Importantly, risk had an additional influence on object choice: with increasing risk difference between objects (ΔRisk, *Equation 8*), choice probability consistently increased for the higher-risk object even with constant value-difference level (*Figure 3D*, yellow and orange bars). The more frequent choice of the riskier object at same value level indicated risk-seeking. A specific logistic regression (*Equation 9*, different from the logistic regression estimating the subjective weights for past trials) confirmed the risk-seeking attitude by a significant positive weight (i.e. beta) of risk on the animals' choices, independent of value (*Figure 3E,F*). When using a subset of trials with minimal value difference for this logistic regression, we again found a significant influence of risk on choices (*Figure 3E*, inset). Formal comparisons favored this choice model based on subjective object value (derived from both weighted reward history and choice history) and subjective object risk over several alternatives, including models without risk, models with different value definitions, models based on objective (true) reward probabilities and risks, and variants of reinforcement learning models (see *Table 1*). Our subjective risk measure also showed specific relationships to saccadic reaction times, alongside value influences on reaction times (*Figure 3—figure supplement 1*). These data confirmed the previously observed positive attitudes of macaques towards objective risk (*Genest et al., 2016*; *Lak et al., 2014*; *McCoy and Platt, 2005*; *O'Neill and Schultz, 2010*; *Stauffer et al., 2014*) and validated our subjective, experience-based object-risk measure as regressor for neuronal activity.

## Neuronal coding of subjective risk associated with choice objects

The subjective variance-risk for specific choice objects, derived from reward history, was coded in 95 of the 205 task-related DLPFC neurons (46%; p<0.05 for object-risk regressors, multiple linear

**Table 1.** Comparison of different models fitted to the animals' choices.
Best fitting model indicated in bold.

| Model | Description | Both animals | | Animal A | | Animal B | |
|-------|-------------|------|------|------|------|------|------|
| | | AIC | BIC | AIC | BIC | AIC | BIC |
| (1) | Value from reward history[1] | 2.2482 | 2.2490 | 1.5077 | 1.5084 | 7.3571 | 7.3636 |
| (2) | Value from reward history and risk[2] | 2.2477 | 2.2492 | 1.5077 | 1.5092 | 7.3522 | 7.3653 |
| (3) | Value from choice history[3] | 2.1614 | 2.1622 | 1.4900 | 1.4907 | 6.5043 | 6.5109 |
| (4) | Value from choice history and risk | 2.0385 | 2.0400 | 1.4023 | 1.4037 | 7.3528 | 7.3660 |
| (5) | Value from reward and choice history[4] | 2.0089 | 2.0097 | 1.3914 | 1.3922 | 6.0880 | 6.0945 |
| **(6)** | **Value from reward and choice history and risk** | **2.0073** | **2.0088** | **1.3899** | **1.3914** | **6.0747** | **6.0878** |
| (7) | Objective reward probabilities[5] | 2.1213 | 2.1220 | 1.4615 | 1.4622 | 6.4972 | 6.5037 |
| (8) | Objective reward probabilities and objective risk[6] | 2.1210 | 2.1225 | 1.4616 | 1.4631 | 6.4982 | 6.5114 |
| (9) | Reinforcement learning (RL) model[7] | 2.0763 | 2.0779 | 1.4376 | 1.4391 | 6.2161 | 6.2293 |
| (10) | RL learning, stack parameter (*Huh et al., 2009*)[8] | 2.0810 | 2.0826 | 1.4374 | 1.4389 | 6.3198 | 6.3330 |
| (11) | RL, reversal-learning variant[9] | 2.2614 | 2.2630 | 1.5330 | 1.5344 | 7.2808 | 7.2939 |

1:Value defined according to *Equation 6*; 2: Risk defined according to *Equation 8*; 3: Value defined as sum of weighted choice history derived from *Equation 5*; 4: Value defined according to *Equation 7*; 5: Objective reward probabilities defined according to *Equation 1*; 6: Objective reward risk defined according to *Equation 2*; 7: Standard Rescorla-Wagner RL model updating value of chosen option based on last outcome; 8: Modified RL model incorporating choice-dependency; 9: Modified RL model updating value of chosen and unchosen option based on last outcome.
DOI: https://doi.org/10.7554/eLife.44838.010

regression, *Equation 10*). These 95 neurons showed 153 object-risk-related responses (among 1222 task-related responses, 13%; 28 of the 153 responses coded risk for both objects, and 125 responses only for one object). Importantly, object-risk coding in these neurons was not explained by object value, which was included as covariate in the regression (shared variance between risk and value regressors: $R^2 = 0.148$ across sessions). A distinct object-choice regressor significantly improved the regression for only 12 responses (p<0.05, partial F-test, *Equation 10*), suggesting most object-risk responses (141/153, 92%) were choice-independent (p=$1.8 \times 10^{-25}$, z-test). A subset of 66 of 153 risk responses (43%) fulfilled our strictest criteria for coding object risk: they coded risk before choice, only for one object, and irrespective of the actual choice, thus complying with requirements for coding a decision variable analogous to object value.

The activity of the neuron in *Figure 4A–B* illustrates the response pattern of an object-risk neuron. During fixation, the activity of the neuron in *Figure 4A* reflected the current risk estimate for object A (p=0.0316, multiple linear regression, *Equation 10*), but was not significant for object-B risk (p=0.69), nor for object values (both p>0.22). True to the concept of a decision input, the risk signal occurred well before the monkey made its choice (in time to inform decision-making) and it was not explained by current-trial choice, cue position, or action (all p>0.46). Classification based on the angle of regression coefficients confirmed coding of object risk, rather than relative risk (*Figure 4C*). Thus, the neuron's activity signaled the continually evolving subjective risk estimate for a specific choice object and may constitute a suitable input for decision mechanisms under risk.

Classification of neuronal responses based on the angle of regression coefficients (*Equation 4*) showed 159 significant risk responses in 80 neurons (p<0.05, F-test), of which 83 responses (52%, 53 neurons, *Figure 4C*) coded object risk rather than risk difference or risk sum. This result confirmed that a substantial number of neurons encoded risk for specific objects; in addition, other neurons encoded risk signals related to both objects as risk difference or risk sum, similar to encoding of value difference and value sum in previous studies (*Cai et al., 2011*; *Tsutsui et al., 2016*; *Wang et al., 2013*). Object risk signals occurred with high prevalence in early trial epochs, timed to potentially inform decision-making (*Figure 4D*). They were recorded in upper and lower principal sulcus banks, confirmed by histology (*Figure 4—figure supplement 1*).

Activity of these object-risk neurons conformed to key patterns that distinguish risk-coding from value-coding. Their population activity followed the typical inverted-U relationship with reward value (cf. *Figure 1A*) by increasing as a function of value within the low-value range and decreasing with value within the high-value range (*Figure 4E*). Accordingly, reward outcomes should increase or decrease risk-related activity depending on whether the additional reward increased or decreased the variance of recent rewards. This feature of reward-variance risk was implicit in the formulation of our risk regressor (*Figure 3C*, magenta curve) and is also illustrated in the population activity in *Figure 4F*. Following reward receipt on trial N-1 (blue curves), activity of risk-neurons on the subsequent trial increased only when the reward led to an increase in reward variance (*Figure 4F* magenta curve, cf. *Equation 8*; ascending slope in *Figure 4E*). By contrast, when reward receipt led to decreased reward variance, neuronal activity on the subsequent trial also decreased (*Figure 4F* green curve; descending slope in *Figure 4E*). Thus, activity of risk neurons followed the evolving statistical variance of rewards, rather than reward probability.

## Control analyses for subjective object-risk coding

Further controls confirmed subjective object-risk coding in DLPFC. Sliding-window regression without pre-selecting responses for task-relatedness identified similar numbers of object-risk neurons as our main fixed-window analysis (82/205 neurons, 40%; *Equation 11*). This sliding-window regression also confirmed that object-risk signals were not explained by past-trial reward, choice, or reward ×choice history, which were regression covariates.

Because our behavioral model estimated risk over multiple past trials, we also tested whether history variables from the past two trials could explain object-risk coding (*Equation 12*). An extended regression identified some responses that reflected nonlinear interactions between rewards over two consecutive past trials (*Figure 4—figure supplement 2A*); these responses might contribute to risk estimation by detecting changes in reward rate. However, they were rare and did not explain our main finding of object-risk coding (*Figure 4—figure supplement 2B*; 99 risk neurons from *Equation 12*).

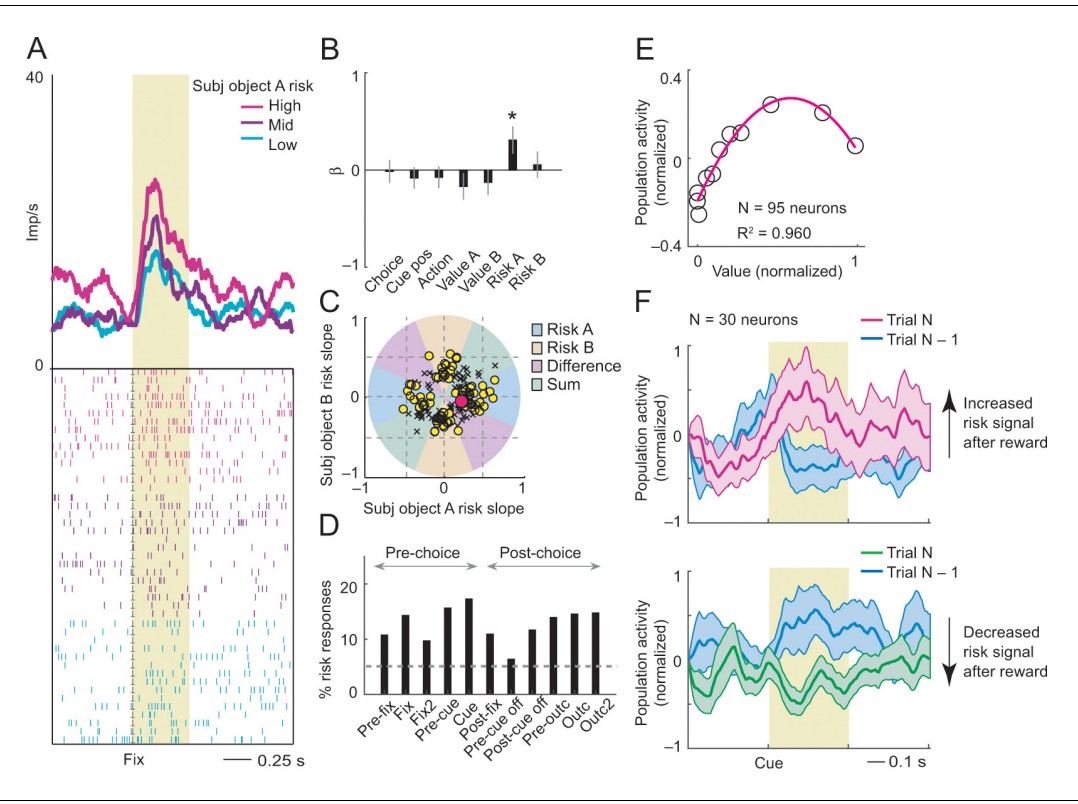

**Figure 4.** Subjective object-risk coding. (**A**) Activity from a single DLPFC neuron coding subjective risk associated with object A before choice (fixation period). Object risk was derived from the variance of recently experienced rewards associated with a specific object. (**B**) Beta coefficients (standardized slopes) from a multiple linear regression of the neuron's fixation impulse rate showed significant coding only for object-A risk (p=0.0316, t-test; all other coefficients: p>0.22). (**C**) Categorization of coding risk for object A or B, risk difference or risk sum based on the angle of coefficients. Red circle: position of object-A risk response of the neuron shown in A and B. (**D**) Percentages of object-risk responses for all task epochs (multiple regression, 1222 task-related responses from 205 neurons). (**E**) Population activity of object-risk neurons as a function of object value. Activity conformed to the characteristic inverted U-shaped relationship between reward-variance risk and reward probability (see *Figure 1A*). Error bars were smaller than symbols. (**F**) Neuronal risk-updating following reward. Population activity of object-risk neurons at the time of choice (cue period), shown separately for trials in which object risk on the current trial increased (top) or decreased (bottom) following reward on the previous trial. When a reward increased object risk (by increasing reward variance), cue-activity on the following trial (N, magenta) was significantly higher compared to that on the previous trial (N-1, blue; p<0.001, Wilcoxon test), reflecting the updated object risk. Conversely, when a reward decreased object risk (by decreasing reward variance), activity on the following trial (green) decreased correspondingly (p<0.001).

DOI: https://doi.org/10.7554/eLife.44838.011

The following source data and figure supplements are available for figure 4:

**Source data 1.**
DOI: https://doi.org/10.7554/eLife.44838.019
**Figure supplement 1.** Anatomical location of recording sites.
DOI: https://doi.org/10.7554/eLife.44838.012
**Figure supplement 1—source data 1.**
DOI: https://doi.org/10.7554/eLife.44838.013
**Figure supplement 2.** Reward-history control.
DOI: https://doi.org/10.7554/eLife.44838.014
**Figure supplement 2—source data 1.**
DOI: https://doi.org/10.7554/eLife.44838.015
**Figure supplement 3.** Control analyses for neuronal object-risk coding.
DOI: https://doi.org/10.7554/eLife.44838.016
*Figure 4 continued on next page*

*Figure 4 continued*

**Figure supplement 3—source data 1.**
DOI: https://doi.org/10.7554/eLife.44838.018
**Figure supplement 4.** Numbers of neurons (and percentages of recorded neurons) encoding risk and value for alternative risk definitions.
DOI: https://doi.org/10.7554/eLife.44838.017

Varying the integration-time windows for risk estimation (using different exponentials and integration up to 15 past trials) resulted in some variation of identified numbers of risk neurons but did not affect our main finding of risk-coding in DLPFC neurons (*Figure 4—figure supplement 3A*).

A direct comparison of objective and subjective risk showed that neuronal activity tended to be better explained by subjective risk. We compared the amount of variance explained by both risk measures when fitting separate regressions. The distributions of partial-$R^2$ values were significantly different between risk measures (*Figure 4—figure supplement 3B*, p=0.0015, Kolmogorov-Smirnov test), attesting to the neuronal separation of these variables. Specifically, subjective risk explained significantly more variance in neuronal responses compared to objective risk (p=0.0406, Wilcoxon test). When both risk measures were included in a stepwise regression model (*Equation 13*), and thus competed to explain variance in neuronal activity, we identified more neurons related to subjective risk than to objective risk (107 compared to 83 neurons, *Figure 4—figure supplement 3C*), of which 101 neurons were exclusively related to subjective risk but not objective risk (shared variance between the two risk measures across sessions: $R^2 = 0.111 \pm 0.004$, mean $\pm$ s.e.m.).

We also considered alternative, more complex definitions of subjective risk that incorporated either weighted reward history or both weighted reward and choice history in the risk calculation. These alternative definitions yielded identical or only slightly higher numbers of identified risk neurons compared to our main risk definition (*Figure 4—figure supplement 4*; less than 5% variation in identified neurons). We therefore focused on our main risk definition (*Equation 8*), which was simpler and more conservative as it incorporated fewer assumptions.

Finally, we examined effects of potential non-stationarity of neuronal activity (*Elber-Dorozko and Loewenstein, 2018*), by including a first-order autoregressive term in *Equation 10*. This resulted in 88 identified risk neurons (compared to 95 neurons in our original analysis). In a further test, we subtracted the activity measured in a control period (at trial start) of the same trial before performing the regression analysis; this procedure should remove effects due to slowly fluctuating neuronal activities. This analysis identified 56 neurons with activity related to risk (note that the control period itself was excluded from this analysis; our original analysis without the control period yielded 81 risk neurons).

Taken together, object-risk signals reflecting our subjective risk measure occurred in significant numbers of DLPFC neurons and were robust to variation in statistical modeling.

## Neuronal coding of subjective risk associated with actions

Neurons in DLPFC process reward values not only for objects (*Tsutsui et al., 2016*) but also for actions (*Khamassi et al., 2015*; *Seo et al., 2012*). Accordingly, we derived subjective action risk from the experienced variance of recent rewards related to specific actions. As our task varied reward probability for particular objects independently of reward probability for particular actions, object risk and action risk showed only low intercorrelation ($R^2 = 0.153$). Among 205 task-related neurons, 90 (44%) coded action risk (148 of 1222 task-related responses, 12%; *Equation 14*). A subset of 77 of 148 action-risk signals (52%) fulfilled our strictest criteria for action risk: they coded risk before the saccadic choice, only for one action, and irrespective of the actual choice, thus complying with requirements for coding a decision variable analogous to action value.

The fixation-period activity of the neuron in *Figure 5A* signaled the risk associated with rightward saccades, reflecting the variance of rewards that resulted from recent rightward saccades (*Figure 5A*; p=0.0233, multiple linear regression, *Equation 14*). The neuronal response was linearly related to risk for rightward but not leftward saccades and failed to correlate with action choice, object choice or action value (all p>0.11; *Figure 5B*). Classification based on the angle of regression coefficients confirmed the designation as an action-risk signal (*Figure 5C*). Thus, the neuron's activity signaled the continually evolving subjective risk estimate for a specific action.

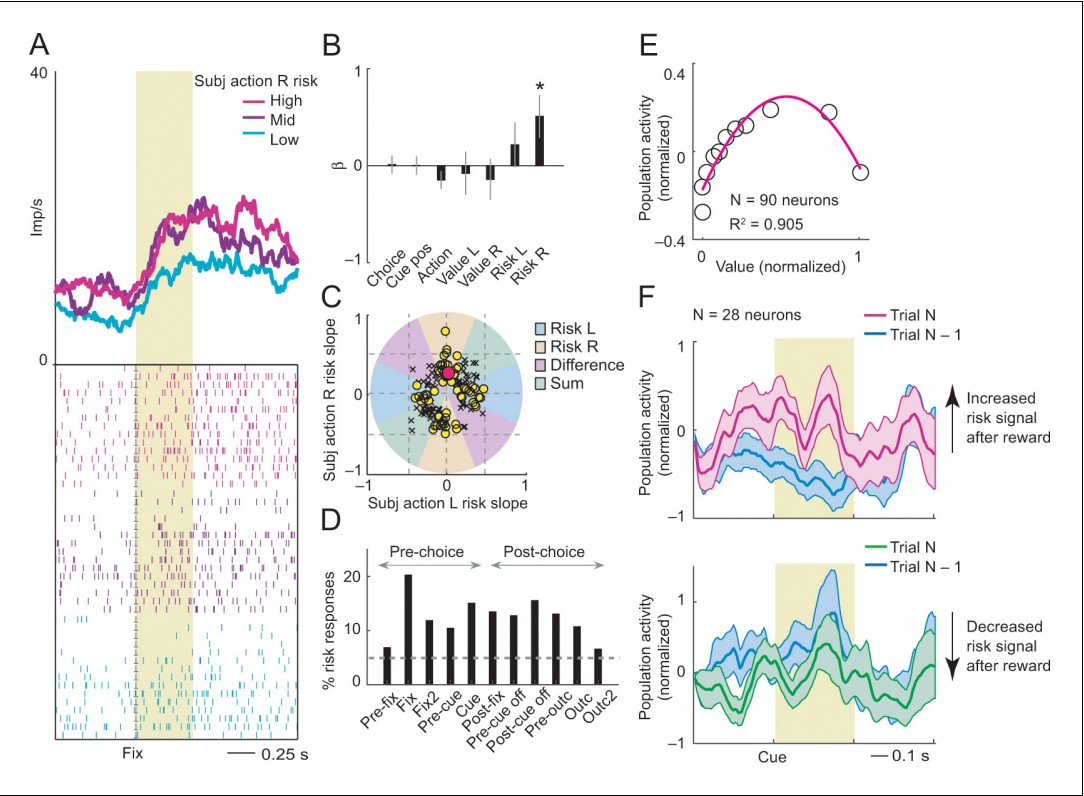

**Figure 5.** Subjective action-risk coding. (**A**) Activity from a single DLPFC neuron coding subjective risk associated with rightward saccades (action R) during fixation. Action risk was derived from the variance of recently experienced rewards associated with a specific action. (**B**) Beta coefficients (standardized slopes) from a multiple linear regression of the neuron's fixation impulse rate showed significant coding only for action risk associated with rightward saccades (p=0.0233, t-test; all other coefficients: p>0.11). (**C**) Categorization of coding risk for action L or R, risk difference or risk sum based on the angle of coefficients. Red circle: position of action-R risk response of the neuron shown in A and B. (**D**) Percentages of action-risk responses for all task epochs (multiple regression, 1222 task-related responses from 205 neurons). (**E**) Population activity of action-risk neurons as a function of reward value. Activity conformed to the characteristic inverted U-shaped relationship between reward-variance risk and reward probability (see *Figure 1A*). Error bars were smaller than symbols. (**F**) Neuronal risk-updating following reward. Population activity of action-risk neurons at the time of choice (cue period), shown separately for trials in which reward on the previous trial increased (top) or decreased (bottom) action risk. When a reward increased action risk (by increasing reward variance), cue-activity on the following trial (N, magenta) was significantly higher compared to that on the previous trial (N-1, blue; p<0.001, Wilcoxon test), reflecting the updated action risk. Conversely, when a reward decreased action risk (by decreasing reward variance), activity on the following trial (green) decreased correspondingly (p<0.001).

DOI: https://doi.org/10.7554/eLife.44838.020

The following source data is available for figure 5:

**Source data 1.**

DOI: https://doi.org/10.7554/eLife.44838.021

Classification of responses based on the angle of regression coefficients (*Equation 4*) showed 149 significant risk responses in 90 neurons (p<0.05, F-test), of which 71 responses (48%, 56 neurons) coded action risk rather than risk difference or risk sum (*Figure 5C*). This result confirmed that a substantial number of neurons encoded risk for specific actions, in addition to neurons encoding risk sum or difference. Action-risk signals occurred throughout all trial periods (*Figure 5D*). Adding an action-choice regressor improved the regression model in only 15 of 148 risk responses (p<0.05, partial F-test). Thus, most responses (133/148, 90%) coded action-risk without additional action-choice coding (p=3.2 × 10$^{-22}$, z-test for dependent samples). Activity of action-risk neurons followed the typical inverted-U relationship with reward value (*Figure 5E*). Reward increased the

activity of these risk-neurons only when the reward increased current reward variance, but decreased neuronal activity when it led to decreased reward variance (*Figure 5F*). Thus, activity of action-risk neurons followed the evolving statistical variance of rewards.

Object risk and action risk were often coded by distinct neurons. Separate multiple regression analyses (*Equations 10 and 14*) revealed that 43 of the 205 task-related neurons (21%) encoded object risk but not action risk, and 38 of the 205 task-related neurons (19%) encoded action risk but not object risk. A stepwise regression on both object risk and action risk (*Equation 15*) resulted in 55 neurons encoding object risk but not action risk (27%) and 38 neurons encoding action risk but not object risk (19%, *Figure 4—figure supplement 3C*). Controlling for non-stationarity of neuronal responses, we identified 83 action-risk neurons when including a first-order autoregressive term and 56 neurons when subtracting neuronal activity at trial start. Neurons encoding object risk and action risk were intermingled without apparent anatomical clustering in DLPFC (*Figure 4—figure supplement 1*), similar to previous studies that failed to detect clustering of object-selective and location-selective neurons (*Everling et al., 2006*).

## Population decoding of object risk and action risk

To quantify the precision with which downstream neurons could read risk information from DLPFC neurons, we used previously validated population-decoding techniques, including nearest-neighbor and linear support-vector-machine classifiers (*Grabenhorst et al., 2012*; *Tsutsui et al., 2016*). We subjected the total of 205 DLPFC neurons to this analysis and did not pre-select risk-neurons for these analyses ('unselected neurons'). We grouped trials according to terciles of object risk and action risk and performed classification based on low vs. high terciles (see Materials and methods).

We successfully decoded object risk and action risk from the population of 205 DLPFC neurons, with accuracies of up to 85% correct in pre-choice trial periods (*Figure 6A,B*). Decoding-accuracy increased as a function of the number of neurons in the decoding sample (*Figure 6A*). Both object risk and action risk were coded with good accuracy across task periods, although object risk was coded significantly more accurately than action risk in several periods (*Figure 6B*). Decoding from randomly sampled small subsets of neurons (N = 20 per sample) showed that risk-decoding accuracy depended on individual neurons' risk sensitivities (standardized regression slopes; *Figure 6C*). Decoding from specifically defined subsets showed that even small numbers of individually significant risk neurons enabled accurate risk-decoding (*Figure 6D*). However, individually significant object-risk neurons carried little information about action risk and individually significant action-risk neurons carried little information about object risk (*Figure 6D*), attesting to the neuronal separation of object risk and action risk in DLPFC. Decoding of risk from neuronal responses remained significantly above chance in control analyses in which we held constant the value of other task-related variables including object choice, action and cue position (*Figure 6E*).

Taken together, unselected population activity carried accurate codes for object risk and action risk. These neuronal population codes depended on population size and individual neurons' risk sensitivities.

## Dynamic integration of risk with reward history, value and choice in single neurons

Neurons often signaled object risk irrespective of other factors. However, over the course of a trial, many neurons dynamically integrated risk with behaviorally important variables in specific ways that were predicted on theoretical grounds. We assessed these coding dynamics with a sliding-window regression (*Equation 11*), which also served to confirm the robustness of our main fixed-window analysis reported above.

Our main, subjective risk measure derived from the history of recently received rewards (*Equation 8*). Following this concept, neurons often combined object-risk signals with information about rewards or choices from previous trials ('history' variables). Early on in trials, the neuron in *Figure 7A* signaled whether or not reward had been received on the last trial, following the choice of a particular object. This signal was immediately followed by an explicit object-risk signal, reflecting the updated, current-trial risk level given the outcome of the preceding trial. In total, 44 neurons showed such joint coding of reward-choice history variables and explicit object risk (54% of 82 risk-coding neurons from sliding-window regression, 21% of 205 recorded neurons; *Equation 11*; *Figure 7B*). By

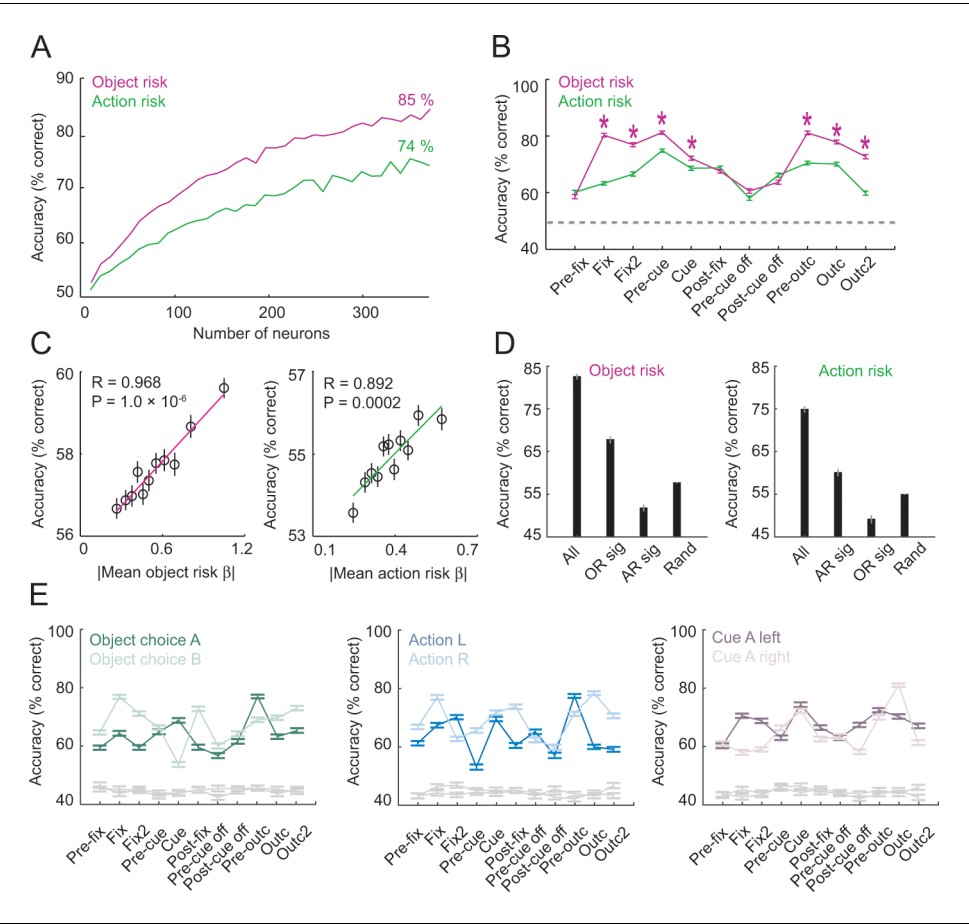

**Figure 6.** Population decoding of risk from unselected neurons. (**A**) Leave-one-out cross-validated decoding accuracy (% correct classification) of a linear support-vector-machine classifier decoding object risk and action risk in pre-cue period. Decoding performance increased with the number of neurons. Analysis was based on normalized single-trial activity of neurons that met decoding criteria without pre-selection for risk-coding. Data for each neuron number show mean (± s.e.m) over 100 iterations of randomly selected neurons. Each neuron entered the decoder twice, to decode risk for object A and B (or action L and R). (**B**) Decoding for object risk and action was significantly above chance (gray line, decoding from shuffled data) in all task epochs (p<0.0001, Wilcoxon test). Object-risk decoding was significantly better than action-risk decoding in specific trial periods (Asterisks: p<0.005, Wilcoxon test). (**C**) Object-risk and action-risk decoding depended on individual neurons' sensitivities for object risk and action risk. Linear regressions of decoding performance in pre-cue period from 5000 subsets of 20 randomly selected neurons on each subset's mean single-neuron regression betas for object risk. (**D**) Decoding accuracy for specific subsets of neurons in pre-cue period. All: entire population of neurons. OR sig: significant object-risk neurons (N = 12). AR sig: significant action-risk neurons (N = 9). Rand: randomly selected neurons (mean ± s.e.m over 5000 randomly selected subsets of N = 20 neurons). (**E**) Decoding accuracy for object-risk decoding from trial subsets in which control variables were held constant. Left to right: decoding object-risk with constant object choice, action and cue position. Decoding was significant for all control variables and all task epochs (p<0.005, Wilcoxon test).

DOI: https://doi.org/10.7554/eLife.44838.022

The following source data is available for figure 6:

**Source data 1.**

DOI: https://doi.org/10.7554/eLife.44838.023

dynamically coding information about recent rewards alongside explicit object-risk signals, these DLPFC neurons seemed suited to contribute to internal risk calculation from experience.

According to the mean-variance approach of finance theory (*D'Acremont and Bossaerts, 2008*; *Markowitz, 1952*), the integration of expected value and risk into utility is thought to underlie

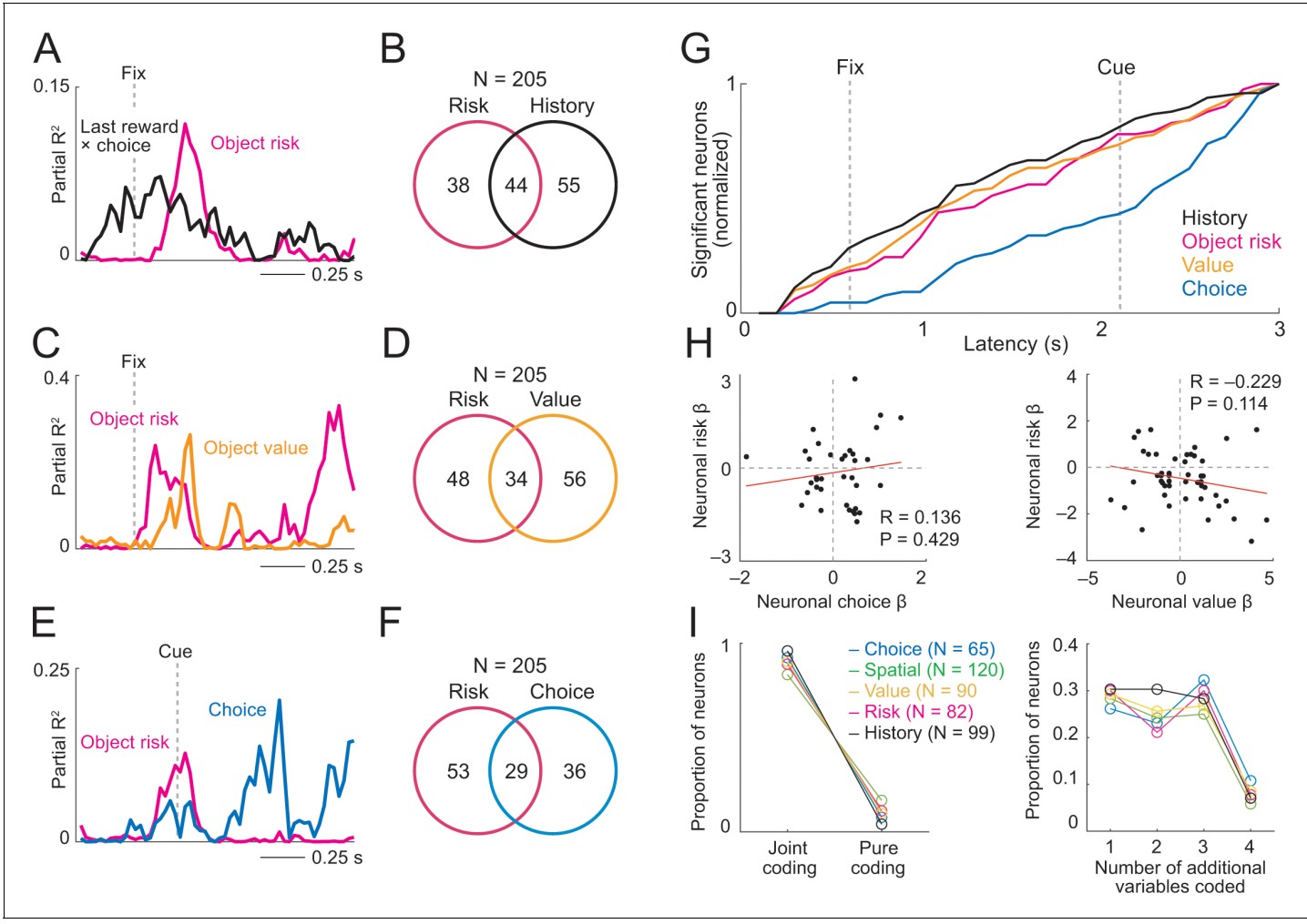

**Figure 7.** Prefrontal neurons dynamically code risk with other behaviorally important variables. (**A**) Neuronal reward history-to-risk transition. A single DLPFC neuron with fixation-period activity that initially reflected whether reward was received from a particular choice on the last trial ('Last reward × choice') before reflecting current-trial object risk. Coefficients of partial determination (partial R²) obtained from sliding-window multiple regression analysis. The observed single-neuron transition from recent reward history to current object risk is consistent with the behavioral model in *Figure 3C* which constructs and updates object-risk estimates from reward experience. (**B**) Numbers of neurons with joint and separate coding of object risk and history variables that were relevant for risk updating (including last-trial reward, last-trial choice, last-trial reward × last trial choice). Numbers derived from sliding window analyses focused on early trial periods relevant to decision-making (trial start until 500 ms post-cue). (**C**) Neuronal value-to-risk integration. A single DLPFC neuron with fixation-period activity encoding both object risk and object value, compatible with the notion of integrating risk and object value into economic utility. (**D**) Number of neurons with joint and separate coding of risk and value. (**E**) Neuronal risk-to-choice transition. A single DLPFC neuron with activity encoding object risk before coding object choice, consistent with decision-making informed by risk. (**F**) Numbers of neurons with joint and separate coding of object risk and object choice. (**G**) Cumulative coding latencies of history variables, object risk, object value, and object choice. Latencies derived from sliding-window regression (first sliding window for which criterion for statistical significance was achieved, see Materials and methods; cumulative proportion of significant neurons normalized to maximum value for each variable). (**H**) Regression coefficients for neurons with joint risk coding and choice coding (left) and joint risk coding and value coding (right). (**I**) Proportion of neurons with joint coding and pure coding of specific task-related variables (left) and proportion of neurons coding different numbers of additional variables (right).
DOI: https://doi.org/10.7554/eLife.44838.024

The following source data and figure supplements are available for figure 7:

**Source data 1.**
DOI: https://doi.org/10.7554/eLife.44838.029
**Figure supplement 1.** Coding of risk and value jointly with spatial variables.
DOI: https://doi.org/10.7554/eLife.44838.025
**Figure supplement 1—source data 1.**
DOI: https://doi.org/10.7554/eLife.44838.026
**Figure supplement 2.** Utility control.
*Figure 7 continued on next page*

*Figure 7 continued*

DOI: https://doi.org/10.7554/eLife.44838.027

**Figure supplement 2—source data 1.**

DOI: https://doi.org/10.7554/eLife.44838.028

behavioral preferences. In agreement with this basic concept, some DLPFC neurons dynamically combined object-risk signals with object-value signals (34 neurons, 41% of 82 risk-coding neurons, 17% of 205 neurons; *Equation 11*; *Figure 7C,D*). The neuron in *Figure 7C* showed overlapping object-value and object-risk signals early in trials during the fixation period, in time to inform object choice on the current trial. This result is potentially consistent with the integration of risk and value into utility. Supplementary analyses (*Figure 7—figure supplement 2*, described below) provided further evidence that some individual neurons integrated risk and value into utility-like signals (although formal confirmation of utility coding would require additional behavioral testing).

If neuronal risk signals in DLPFC contributed to decisions, the activity of individual neurons might reflect the forward information flow predicted by computational decisions models (*Deco et al., 2013*; *Grabenhorst et al., 2019*; *Wang, 2008*), whereby reward and risk evaluations precede choice. Thus, object risk as an important decision variable and the resulting, subsequent object choice should be jointly represented by neurons during decision-making. Indeed, activity in some DLPFC neurons dynamically combined object risk with the choice the animal was going to make (29 neurons, 35% of 82 risk-coding neurons, 14% of 205 recorded neurons; *Equation 11*; *Figure 7E,F*). At the time of choice, the neuron in *Figure 7E* signaled the risk of a specific object moments before it signaled the object choice for that trial, consistent with theoretically predicted transformations of risk and value signals into choice. Comparing the coding latencies for different variables across the population of DLPFC neurons, signals for reward history, object risk and object value arose significantly earlier than choice signals (p<0.0001, rank-sum tests; *Figure 7G*).

The percentages of neurons coding specific pairs of variables was not significantly different than expected given the probabilities of neurons coding each individual variable (history and risk: $\chi 2 = 1.58$, p=0.2094, value and risk: $\chi 2 = 3.54$, p=0.0599, choice and risk: $\chi 2 = 0.845$, p=0.358). We also tested for relationships in the coding scheme (measured by signed regression coefficients) among neurons with joint risk and choice coding or joint risk and value coding. Across neurons, there was no significant relationship between the regression coefficients (standardized slopes) for the different variables (*Figure 7H*). This suggested that while some neurons used corresponding coding schemes for these variables (risk and choice, risk and value) other neurons used opposing coding schemes (see Discussion).

Overall, the majority of DLPFC neurons coded task-related variables in combination with other variables (*Figure 7I*). Thus, 'pure' coding of any given variable, including object risk, was rare in DLPFC and many neurons dynamically combined these signals with one or more additional variables (z-tests for dependent samples comparing proportion of joint and pure coding: $p<1.6 \times 10^{-13}$ for all variables in *Figure 7I*). In addition to the risk-related dynamic coding transitions described above, activity in some DLPFC neurons transitioned from coding risk to coding of spatial variables such as cue position or action choice (*Figure 7—figure supplement 1*).

Thus, in addition to pure risk coding, DLPFC neurons frequently combined object risk with other reward and decision parameters on individual trials. These neurons provide a suitable basis for internal risk calculation and for the influence of risk on economic choices.

## Utility control

According to approaches in finance theory, risk is integrated with expected value into utility (*D'Acremont and Bossaerts, 2008*; *Markowitz, 1952*). We tested whether neuronal risk responses were accounted for by subjective integration of value (derived from the mean of the recent reward and choice histories) and risk (derived from the variance of the recent reward history). We calculated this mean-variance utility as a weighted sum of object value and risk with the weights for value and risk derived from logistic regression (*Equation 9*).

The neuron in *Figure 7—figure supplement 2A* reflected the utility of object A (p=0.025; *Equation 10*, with object-utility regressors substituting object-value regressors); it failed to reflect utility

of object B, object risk, cue position or action (*Figure 7—figure supplement 2B*, all p>0.075) but reflected in addition the object choice (p=0.0029). Multiple regression identified such responses reflecting the utility of individual objects (*Figure 7—figure supplement 2C,D*, *Equation 10*). Specifically, 109 responses (97 neurons) were classified as coding utility (based on the angle of utility coefficients, *Equation 4*). Population decoding accuracy for utility was significant across task periods (*Figure 7—figure supplement 2E*). As for risk, decoding accuracy increased with more neurons and dependent on individual neurons' sensitivities (*Figure 7—figure supplement 2F,G*).

Neuronal coding of utility did not account for our main finding of risk-coding. Regression with utility and risk regressors (in addition to object choice, cue position and action regressors) showed 108 neurons with significant risk responses (52.7% of neurons). Among them, 34 neurons were significant for risk but not utility. A stepwise regression resulted in 222 risk responses (of 1222 task-related responses, 18%) and 186 utility responses (15.2%). Thus, risk-related signals were not accounted for by utility.

## Discussion

These data suggest that neurons in primate DLPFC signal the variance of fluctuating rewards derived from recent experience. Neuronal risk signals correlated with the subjective risk estimated from the animal's choices and with experimentally programmed, objective risk. The frequently changing risk levels in the choice task required the animal to derive risk from the statistics of experienced rewards, rather than from cues that explicitly signaled risk levels (such as pre-trained risk-associated bar stimuli or fractals), in order to make choices. The variance-tracking neurons encoded risk information explicitly, and distinctly, as object risk and action risk; these risk signals were specific for particular objects or actions, occurred before the animal's choice, and typically showed little dependence on the object or action being chosen, thus complying with criteria for decision inputs (*Sutton and Barto, 1998*). Some neurons dynamically combined risk with information about past rewards, current object values or future choices, and showed transitions between these codes within individual trials; these characteristics are consistent with theoretical predictions of decision models (*Deco et al., 2013*; *Grabenhorst et al., 2019*; *Wang, 2008*) and our model of subjective risk estimation from recent experience (*Figure 3C*). These prefrontal risk signals seem to provide important information about dynamically evolving risk estimates as crucial components of economic decisions under uncertainty.

Risk is an abstract variable that is not readily sensed by decision-makers but requires construction from experience or description (*Hertwig et al., 2004*). Reward probabilities in our choice task varied continually, encouraging the animals to alternately choose different objects and actions. Under such conditions, reward value and risk for objects and actions vary naturally with the animals' behavior. We followed previous studies that estimated subjective reward values, rather than objective physical values, as explanatory variables for behavior and neurons (*Lau and Glimcher, 2005*; *Lau and Glimcher, 2008*; *Samejima et al., 2005*; *Sugrue et al., 2004*). To estimate subjective risk, we adapted an established approach for reward-value estimation based on the integration of rewards over a limited window of recent experiences (*Lau and Glimcher, 2005*). We used this approach to calculate subjective risk from the variance of recent reward experiences and showed that the animals' choices (*Figure 3D–F*) and prefrontal neurons (*Figures 4* and *5*) were sensitive to these continually evolving subjective risk estimates, irrespective of value.

We could detect risk neurons also with an objective risk measure derived from true reward probability. However, our subjective risk measure took into account that the animals likely had imperfect knowledge of true reward probabilities and imperfect memory for past rewards. Accordingly, the observed neuronal relationships to objective risk are unlikely to reflect perfect tracking of the environment by the neurons; rather, the correlation with objective risk is likely explained by the fact that objective and subjective risk were related. This situation is similar to previous studies that compared coding of objective and subjective values (*Lau and Glimcher, 2008*; *Samejima et al., 2005*).

Our definition of subjective risk has some limitations. To facilitate comparisons with previous studies, we restricted our definition to the variance of past reward outcomes; we did not extend the risk measure to the variance of past choices, which would not have a clear correspondence in the economic or neuroeconomic literature. Our subjective risk measure provided a reasonable account of behavioral and neuronal data: it had a distinct relationship to choices, was encoded by a substantial

number of neurons, and tended to better explain neuronal responses compared to objective risk. We followed the common approach of calculating reward statistics over a fixed temporal window because of its generality and simplicity, and to link our data to previous studies. An extension of this approach could introduce calculation of reward statistics over flexible time windows, possibly depending on the number of times an option was recently chosen (similar to an adaptive learning rate in reinforcement learning models).

Many neurons encoded risk for specific objects or actions prior to choice and independently of choice. Such pure risk signals could provide inputs for competitive, winner-take-all decision mechanisms that operate on separate inputs for different options (*Deco et al., 2013*; *Grabenhorst et al., 2019*; *Wang, 2008*). The presence of DLPFC neurons that transitioned from risk-coding to choice-coding (*Figure 7E*) is consistent with this interpretation. In addition to providing pure risk inputs to decision-making, object-risk signals could converge with separately coded value signals to inform utility calculations, which require integration of risk with expected value according to individual risk attitude (*D'Acremont and Bossaerts, 2008*; *Markowitz, 1952*). This possibility is supported by DLPFC neurons that dynamically encoded both risk and value for specific choice objects (*Figure 7C*). However, confirmation that DLPFC neurons encode utility will require further experimental testing with formal utility curves (*Genest et al., 2016*; *Stauffer et al., 2014*). Thus, risk neurons in DLFPC seem well suited to contribute to economic decisions, either directly by providing risk-specific decision inputs or indirectly by informing utility calculations.

Notably, while some DLPFC neurons jointly coded risk with value or choice in a common coding scheme (indicated by regression coefficients of equal sign), this was not the rule across all neurons with joint coding (*Figure 7H*). This result and the observed high degree of joint coding, with most DLPFC dynamically coding several task-related variables (*Figure 7I*), matches well with previous reports that neurons in DLPFC show heterogeneous coding and mixed selectivity (*Rigotti et al., 2013*; *Wallis and Kennerley, 2010*). An implication for the present study might be that risk signals in DLPFC can support multiple cognitive processes in addition to decision-making, as also suggested by the observed relationship between risk and reaction times (*Figure 3—figure supplement 1*).

Objects (or goods) represent the fundamental unit of choice in economics (*Padoa-Schioppa, 2011*; *Schultz, 2015*), whereas reinforcement learning in machine learning conceptualizes choice in terms of actions (*Sutton and Barto, 1998*). The presently observed neuronal separation of object risk and action risk is analogous to neuronal separation of value signals for objects (*Grabenhorst et al., 2019*; *Padoa-Schioppa, 2011*; *So and Stuphorn, 2010*; *Tsutsui et al., 2016*) and actions (*Lau and Glimcher, 2008*; *Samejima et al., 2005*; *Seo et al., 2012*). Accordingly, object-risk and action-risk signals could provide inputs to separate mechanisms contributing to the selection of competing objects and actions. Additionally observed neuronal signals related to the sum or difference of object risk could contribute separately to decision-making and motivation processes, similar to previously observed neuronal coding of value sum and value difference (*Cai et al., 2011*; *Tsutsui et al., 2016*; *Wang et al., 2013*).

Previous studies reported risk-sensitive neurons in dopaminergic midbrain (*Fiorillo et al., 2003*; *Lak et al., 2014*; *Stauffer et al., 2014*), orbitofrontal cortex (*O'Neill and Schultz, 2010*; *O'Neill and Schultz, 2013*; *Raghuraman and Padoa-Schioppa, 2014*), cingulate cortex (*McCoy and Platt, 2005*; *Monosov, 2017*), basal forebrain (*Ledbetter et al., 2016*; *Monosov and Hikosaka, 2013*), and striatum (*White and Monosov, 2016*). In one study, orbitofrontal neurons encoded 'offer risk' for specific juice types (*Raghuraman and Padoa-Schioppa, 2014*), analogous to the presently observed risk signals for specific visual objects. Critically, most previous studies used explicit risk-descriptive cues indicating fixed risk levels and tested neuronal activity when the animals had already learned cue-associated risk levels. One series of studies (*Monosov and Hikosaka, 2013*; *White and Monosov, 2016*) documented how risk responses evolved for novel cues with fixed risk levels, although relations to statistical variance of reward history were not examined. Here, we examined how neuronal risk estimates are derived internally and continually updated based on distinct reward experiences.

How could variance estimates be computed neurally? In neurophysiological models, reward signals modify synaptic strengths of valuation neurons, with decaying influences for more temporally remote rewards (*Wang, 2008*). Variance-risk could be derived from such neurons by a mechanism that registers deviations of reward-outcomes from synaptically stored mean values. This process may involve risk-sensitive prediction error signals in dopamine neurons (*Lak et al., 2014*; *Stauffer et al.,*

2014), orbitofrontal cortex (*O'Neill and Schultz, 2013*) and insula (*Preuschoff et al., 2008*). Our data cannot determine whether DLPFC neurons perform such variance computation or whether the observed risk signals reflected processing elsewhere; resolving this question will require simultaneous recordings from multiple structures. Nevertheless, a role of DLPFC neurons in local risk computation would be consistent with known prefrontal involvement in temporal reward integration (*Barraclough et al., 2004*; *Seo et al., 2007*), including recently described integration dependent on volatility (*Massi et al., 2018*), processing of numerical quantities and mathematical rules (*Bongard and Nieder, 2010*; *Nieder, 2013*), and the currently observed transitions from past reward coding to object risk coding (*Figure 7A,B*). Moreover, in one recent study, reward information from past trials enhanced the encoding of current-trial task-relevant information in DLPFC neurons (*Donahue and Lee, 2015*).

The prefrontal cortex has long been implicated in adaptive behavior (*Miller and Cohen, 2001*), although economic risk processing is often associated with its orbital part, rather than the dorsolateral region studied here (*O'Neill and Schultz, 2010*; *Padoa-Schioppa, 2011*; *Stolyarova and Izquierdo, 2017*). However, DLPFC neurons are well suited to signal object risk and action risk based on recent reward variance: DLPFC neurons process numerical quantities and basic mathematical rules (*Bongard and Nieder, 2010*; *Nieder, 2013*), integrate reward information over time (*Barraclough et al., 2004*; *Donahue and Lee, 2015*; *Massi et al., 2018*; *Seo et al., 2007*), and process both visual objects and actions (*Funahashi, 2013*; *Suzuki and Gottlieb, 2013*; *Watanabe, 1996*). Previous studies implicated DLPFC neurons in reward valuation during decision-making (*Cai and Padoa-Schioppa, 2014*; *Hosokawa et al., 2013*; *Kennerley et al., 2009*; *Kim and Shadlen, 1999*; *Wallis and Miller, 2003*). We recently showed that DLPFC neurons encoded object-specific values and their conversion to choices (*Tsutsui et al., 2016*). A previous imaging study detected a risk-dependent reward value signal in human lateral prefrontal cortex but no separate, value-independent risk signal (*Tobler et al., 2009*), perhaps due to insufficient spatiotemporal resolution. Importantly, the presently described risk signals were not explained by value, which we controlled for in our regressions. Thus, the presently observed DLPFC risk neurons may contribute to economic decisions beyond separately coded value signals.

Activity in DLPFC has been implicated in attention (*Everling et al., 2002*; *Squire et al., 2013*; *Suzuki and Gottlieb, 2013*) and the presently observed risk signals may contribute to DLPFC's object and spatial attentional functions (*Miller and Cohen, 2001*; *Watanabe, 1996*). However, several observations argue against an interpretation of our results solely in terms of attention. First, DLPFC neurons encoded risk with both positive and negative slopes, suggesting no simple relationship to attention, which is usually associated with activity increases (*Beck and Kastner, 2009*; *Hopfinger et al., 2000*; *Squire et al., 2013*). Second, in many neurons, risk signals were dynamically combined with signals related to other task-relevant variables, suggesting specific functions in risk updating and decision-making. Thus, the present neuronal risk signals may contribute to established DLPFC functions in attention (*Everling et al., 2002*; *Squire et al., 2013*; *Suzuki and Gottlieb, 2013*) but also seem to play distinct, more specific roles in decision processes.

Our findings are consistent with a potential role for DLPFC neurons in signaling economic value and risk, and in converting these signals to choices. This general notion is supported by an earlier study implicating DLPFC in conversions from sensory evidence to oculomotor acts (*Kim and Shadlen, 1999*). However, the choices may not be computed in DLPFC. Indeed, previous studies showed that value signals occur late in DLPFC, at least following those in orbitofrontal cortex (*Kennerley et al., 2009*; *Wallis and Miller, 2003*). A recent study using explicitly cued juice rewards demonstrated conversions from chosen juice signals to action signals in DLPFC but apparently few DLPFC neurons encoded the value inputs to these choices (*Cai and Padoa-Schioppa, 2014*). By contrast, risk in our task was derived from integrated reward history, to which DLPFC neurons are sensitive (*Barraclough et al., 2004*; *Massi et al., 2018*; *Seo et al., 2007*). It is possible that DLPFC's involvement in converting decision parameters (including object risk as shown here) to choice signals depends on task requirements. This interpretation is supported by a recent study which showed that the temporal evolution of decision signals in DLPFC differs between delay-based and effort-based choices, and that orbitofrontal and anterior cingulate cortex differentially influence DLPFC decision signals for these different choice types (*Hunt et al., 2015*).

In summary, these results show that prefrontal neurons tracked the evolving statistical variance of recent rewards that resulted from specific object choices and actions. Such variance-sensing neurons

in prefrontal cortex may provide a physiological basis for integrating discrete event experiences and converting them into representations of abstract quantities such as risk. By coding this quantity as object risk and action risk, these prefrontal neurons provide distinct and specific risk inputs to economic decision processes.

## Materials and methods

### Animals

All animal procedures conformed to US National Institutes of Health Guidelines and were approved by the Home Office of the United Kingdom (Home Office Project Licenses PPL 80/2416, PPL 70/8295, PPL 80/1958, PPL 80/1513). The work has been regulated, ethically reviewed and supervised by the following UK and University of Cambridge (UCam) institutions and individuals: UK Home Office, implementing the Animals (Scientific Procedures) Act 1986, Amendment Regulations 2012, and represented by the local UK Home Office Inspector; UK Animals in Science Committee; UCam Animal Welfare and Ethical Review Body (AWERB); UK National Centre for Replacement, Refinement and Reduction of Animal Experiments (NC3Rs); UCam Biomedical Service (UBS) Certificate Holder; UCam Welfare Officer; UCam Governance and Strategy Committee; UCam Named Veterinary Surgeon (NVS); UCam Named Animal Care and Welfare Officer (NACWO).

Two adult male macaque monkeys (*Macaca mulatta*) weighing 5.5–6.5 kg served for the experiments. The animals had no history of participation in previous experiments. The number of animals used and the number of neurons recorded for the experiment is typical for studies in this field of research; we did not perform explicit power analysis. A head holder and recording chamber were fixed to the skull under general anaesthesia and aseptic conditions. We used standard electrophysiological techniques for extracellular recordings from single neurons in the sulcus principalis area of the frontal cortex via stereotaxically oriented vertical tracks, as confirmed by histological reconstruction. After completion of data collection, recording sites were marked with small electrolytic lesions (15–20 μA, 20–60 s). The animals received an overdose of pentobarbital sodium (90 mg/kg iv) and were perfused with 4% paraformaldehyde in 0.1 M phosphate buffer through the left ventricle of the heart. Recording positions were reconstructed from 50-μm-thick, stereotaxically oriented coronal brain sections stained with cresyl violet.

### Behavioral task

The animals performed an oculomotor free-choice task involving choices between two visual objects to each of which reward was independently and stochastically assigned. Trials started with presentation of a red fixation spot (diameter: 0.6°) in the center of a computer monitor (viewing distance: 41 cm; *Figure 1B*). The animal was required to fixate the spot and contact a touch sensitive, immobile resting key. An infrared eye tracking system continuously monitored eye positions (ISCAN, Cambridge, MA). During the fixation period at 1.0–2.0 s after eye fixation and key touch, an alert cue covering the fixation spot appeared for 0.7–1.0 s. At 1.4–2.0 s following offset of the alert cue, two different visual fractal objects (A, B; square, 5° visual angle) appeared simultaneously as ocular choice targets on each side of the fixation spot at 10° lateral to the center of the monitor. Left and right positions of objects A and B alternated pseudorandomly across trials. The animal was required to make a saccadic eye movement to its target of choice within a time window of 0.25–0.75 s. A red peripheral fixation spot replaced the target after 1.0–2.0 s of target fixation. This fixation spot turned to green after 0.5–1.0 s, and the monkey released the touch key immediately after color change. Rewarded trials ended with a fixed quantity of 0.7 ml juice delivered immediately upon key release. A computer-controlled solenoid valve delivered juice reward from a spout in front of the animal's mouth. Unrewarded trials ended at key release and without further stimuli. The fixation requirements restricted eye movements from trial start to cue appearance and, following the animals' saccade choice, from choice acquisition to reward delivery. This ensured that neuronal activity was minimally influenced by oculomotor activity, especially in our main periods of interest before cue appearance.

Reward probabilities of object A and B were independently calculated in every trial, depending on the numbers of consecutive unchosen trials (*Equation 1*):

$$P = 1 - (1 - P_0)^{n+1} \tag{1}$$

with $P$ as instantaneous reward probability, $P_0$ as experimentally imposed, base probability setting, and $n$ as the number of trials that the object had been consecutively unchosen. This equation describes the probabilistic reward schedule of the Matching Law (*Herrnstein, 1961*) in defining how the likelihood of being rewarded on a target increased with the number of trials after the object was last chosen but stayed at base probability while the object was repeatedly chosen (irrespective of whether that choice was rewarded or not). Reward was thus probabilistically assigned to the object in every trial, and once a reward was assigned, it remained available until the associated object was chosen.

We varied the base reward probability in blocks of typically 50–150 trials (chosen randomly) without signaling these changes to the animal. We used different base probabilities from the range of p=0.05 to p=0.55, which we chose randomly for each block. The sum of base reward probabilities for objects A and B was held constant so that only relative reward probability varied. The trial-by-trial reward probabilities for objects A and B, calculated according to *Equation 1* varied within the range of p=0.05 to p=0.99.

## Calculation of objective risk

A most basic, assumption-free definition of objective risk derives risk directly from the variance of the true, specifically set (i.e. programmed) reward probabilities that changed on each trial depending on the animal's matching behavior (*Equation 1*). Accordingly, we used the conventional definition of variance to calculate risk in each trial from the programmed, binary probability distribution (Bernoulli distribution) that governed actual reward delivery in each trial (*Equation 2*):

$$var = \sum_k \left[ p_k(m_k) \times (m_k - EV)^2 \right] \tag{2}$$

with $p$ as the trial-specific probability derived from *Equation 1*, $m$ as reward magnitude (0.7 ml for reward, 0 for no-reward outcome), $k$ as outcome (0 ml or set ml of reward) and $EV$ as expected value (defined as the sum of probability-weighted reward amounts). In our task, reward magnitude on rewarded trials was held constant at 0.7 ml; the definition generalizes to situations with different magnitudes. With magnitude $m$ held constant, variance risk follows an inverted-U function of probability (*Figure 1A*). The risk for objects A and B, calculated as variance according to *Equation 2* varied within the range of var = 0.0003 to var = 0.1225.

## Objective risk: analysis of neuronal data

We counted neuronal impulses in each neuron on correct trials relative to different task events with 500 ms time windows that were fixed across neurons: before fixation spot (Pre-fix, starting 500 ms before fixation onset), early fixation (Fix, following fixation onset), late fixation (Fix2, starting 500 ms after fixation spot onset), pre-cue (Pre-cue, starting 500 ms before cue onset), cue (Cue, following cue onset), post-fixation (Post-fix, following fixation offset), before cue offset (Pre-cue off, starting 500 ms before cue offset), after cue offset (Post-cue off, following cue offset), pre-outcome (Pre-outc, starting 500 ms before reinforcer delivery), outcome (Outc, starting at outcome delivery), late outcome (Outc2, starting 500 ms after outcome onset).

We first identified task-related responses in individual neurons and then used multiple regression analysis to test for different forms of risk-related activity while controlling for the most important behaviorally relevant covariates. We identified task-related responses by comparing activity to a control period (Pre-fix) using the Wilcoxon test (p<0.005, Bonferroni-corrected for multiple comparisons). A neuron was included as task-related if its activity in at least one task period was significantly different to that in the control period. Because the Pre-fixation period served as control period we did not select for task-relatedness in this period and included all neurons with observed impulses in the analysis. We chose the pre-fixation period as control period because it was the earliest period at the start of a trial in which no sensory stimuli were presented. The additional use of a sliding-window regression approach for which no comparison with a control period was performed (see below) confirmed the results of the fixed window analysis that involved testing for task-relationship.

We used multiple regression analysis to assess relationships between neuronal activity and task-related variables. Statistical significance of regression coefficients was determined using t-test with $p < 0.05$ as criterion. Our analysis followed established approaches previously used to test for value coding in different brain structures (*Lau and Glimcher, 2008*; *Samejima et al., 2005*). All tests performed were two-sided.

Each neuronal response was tested with the following multiple regression model to identify responses related to objective, true risk derived from reward probability (*Equation 3*):

$$y = \beta_0 + \beta_1 ObjectChoice + \beta_2 CuePosition + \beta_3 Action + \beta_4 TrueProbA$$
$$+ \beta_5 TrueProbB + \beta_6 TrueRiskA + \beta_7 TrueRiskB + \varepsilon$$

(3)

with $y$ as trial-by-trial neuronal impulse rate, *ObjectChoice* as current-trial object choice (0 for A, one for B), *CuePosition* as current-trial spatial cue position (0 for object A on the left, one for object A on the right), *Action* as current-trial action (0 for left, one for right), *TrueProbA* as the true current-trial reward probability of object A calculated from *Equation 1*, *TrueProbB* as the true current-trial reward probability of object B calculated from *Equation 1*, *TrueRiskA* as the true current-trial risk of object A calculated from *TrueProbA* according to *Equation 2*, *TrueRiskB* as the true current-trial risk of object B calculated from *TrueProbB* according to *Equation 2*, $\beta_1$ to $\beta_7$ as corresponding regression coefficients, $\beta_0$ as constant, $\varepsilon$ as residual. A neuronal response was classified as coding object risk if it had a significant coefficient for *TrueRiskA* or *TrueRiskB*.

We used the same model to test for neuronal coding of objective action risk by substituting the object-risk regressors with action-specific risk regressors (defined by the left-right object arrangement on each trial; thus, if object A appeared on the left side on a given trial, the action L risk regressor would be determined by the object A risk regressor for that trial).

An alternative method for classification of neuronal risk responses used the angle of regression coefficients (*Tsutsui et al., 2016*; *Wang et al., 2013*). This classification method is 'axis-invariant' as it is independent on the axis choice for the regression model, that is whether the model includes separate variables for object risk or separate variables for risk sum and difference (*Wang et al., 2013*). However, the regression in this form omits relevant variables coded by DLPFC neurons (choice, cue position, action); accordingly, we use this approach as additional confirmation for our main regression above. We fitted the following regression model (*Equation 4*):

$$y = \beta_0 + \beta_1 ObjectRiskA + \beta_2 ObjectRiskB + \varepsilon$$

(4)

Using this method, a neuronal response was categorized as risk-related if it showed a significant overall model fit ($p < 0.05$, *F*-test), rather than testing the significance of individual regressors. For responses with significant overall model fit, we plotted the magnitude of the beta coefficients (slopes) of the two object-risk regressors on an x-y plane (*Figure 2C*). We followed a previous study (*Wang et al., 2013*) and divided the coefficient space into eight equally spaced segments of 45° to categorize neuronal responses based on the polar angle. We classified responses as coding object risk ('absolute risk') if their coefficients fell in the segments pointing toward 0° or 180° (object risk A) or toward 90° or 270° (object risk B). We used an analogous procedure for action risk classification. We classified responses as coding risk difference if their coefficients fell in the segments pointing toward 135° or 315° and as coding risk sum if their coefficients fell in the segments pointing toward 45° or 225°.

## Logistic regression for defining the weight of past reward

The animal may have paid more attention to immediately past rewards compared to earlier, more remote rewards. To establish a potential subjective reward value weighing, we used a logistic regression to model subjective reward value from the animals' past rewards and choices, similar to comparable previous behavioral and neurophysiological macaque studies (*Corrado et al., 2005*; *Lau and Glimcher, 2005*; *Sugrue et al., 2004*). As our task involved tracking changing values of objects (one fractal image for each of the two choice options), we formulated the model in terms of object choices rather than action choices. We fitted a logistic regression to the animal's trial-by-trial choice data to estimate beta coefficients for the recent history of received rewards and recently made choices. Note that in choice tasks such as the one used here, choices and reward outcomes depend not only on reward history but also on choice history (*Lau and Glimcher, 2005*). Thus,

rewards and choices were both included into the logistic regression to avoid an omitted variable bias and provide a more accurate estimation of the weighting coefficients. The resulting coefficients quantified the extent to which the animals based their choices on recently received rewards and recently made choices for a given option; thus, the coefficients effectively weighed past trials with respect to their importance for the animal's behavior. We used the following logistic regression to determine the weighting coefficients for reward history ($\beta_j^r$) and choice history ($\beta_j^c$) (*Equation 5*):

$$\log\left(\frac{p_A(i)}{p_B(i)}\right) = \sum_{j=1}^{N}\beta_j^r(R_A(i-j)-R_B(i-j)) + \sum_{j=1}^{N}\beta_j^c(C_A(i-j)-C_B(i-j)) + \beta_0 \tag{5}$$

with $p_A(i)$ [or $p_B(i)$] as the probability of choosing object A (or B) on the $i$th current trial, $R_A$[or $R_B$] as reward delivery after choice of object A [or B] on the $i$th trial, $j$ is the past trial relative to the $i$th trial, $C_A$[or $C_B$] as choice of object A [or B] on the $i$th trial, $N$ denoting the number of past trials included in the model ($N$ = 10), and $\beta_0$ as bias term. Exploratory analysis had shown that the beta coefficients did not differ significantly from 0 for more than ($N$ = 10) past trials. Thus, *Equation 5* modeled the dependence of the monkeys' choices on recent rewards and recent choices for specific objects; by fitting the model we estimated the subjective weights (i.e. betas) that animals placed on recent rewards and choices. Thus, the crucial weighting coefficients reflecting the subjective influence of past rewards and choices were $\beta_j^r$ and $\beta_j^c$. As described below, the $\beta_j^r$ coefficients were also used for calculating the variance that served as the subjective risk measure of our study. The logistic regression was estimated by fitting regressors to a binary indicator function (dummy variable), setting the variable to 0 for the choice of one object and to 1 for the choice of the alternative object, using a binomial distribution with logit link function. The coefficients for reward and choice history from this analysis are plotted in *Figure 3A and B* as reward and choice weights.

## Calculation of weighted, subjective value

To calculate the subjective value of each reward object from the animal's experience, we used the weights estimated from the logistic regression (*Equation 5*). We followed previous studies of matching behavior (*Lau and Glimcher, 2005*) that distinguished two influences on value: the history of recent rewards and the history of recent choices. The first object-value component related to reward history, $OV_A^r$, can be estimated by the mean of subjectively weighted reward history over the past 10 trials (*Equation 6*):

$$OV_A^r = \frac{\sum_{j=1}^{N}\beta_j^r(R_A(i-j))}{N} \tag{6}$$

with $R_A$ again as reward delivery after choice of object A on the $i$th trial, $j$ as the past trial relative to the $i$th trial, $N$ the number of past trials included in the model ($N$ = 10), $\beta_j^r$ as regression coefficient for the weight of past rewards (estimated by *Equation 5*).

In tasks used to study matching behavior, such as the present one, it has been shown that choice history has an additional influence on behavior and that this influence can be estimated using logistic regression (*Lau and Glimcher, 2005*). To account for this second object-value component related to choice history, we estimated a subjective measure of object value that incorporated both a dependence on weighted reward history and a dependence on weighted choice history (*Equation 7*):

$$OV_A^{r,c} = \frac{\sum_{j=1}^{N}\beta_j^r(R_A(i-j)) + \sum_{j=1}^{N}\beta_j^c(C_A(i-j))}{N} \tag{7}$$

with $R_A$ as reward delivery after choice of object A on the $i$th trial, $C_A$[or $C_B$] as choice of object A [or B] on the $i$th trial, $j$ as the past trial relative to the $i$th trial, $N$ the number of past trials included in the model ($N$ = 10), $\beta_j^r$ as regression coefficient for the weight of past rewards and $\beta_j^c$ as regression coefficient for the weight of past choice (estimated by *Equation 5*). This measure of subjective object value (*Equation 7*) based on both weighted reward and choice history, $OV_A^{r,c}$, constituted our main value measure for behavioral and neuronal analyses.

## Calculation of subjective risk

Our aim was to construct a subjective risk measure that derived risk from the variance of recent reward experiences and that incorporated the typically observed decreasing influence of past trials on the animal's behavior. We used the following definition as our main measure of subjective object risk (*Equation 8*):

$$var_A = \frac{\sum_{j=1}^{N} \beta_j^r \left( R_A(i-j) - \left( \sum_{j=1}^{N} (R_A(i-j)) \right) /N \right)^2}{N-1} \tag{8}$$

with $\beta_j^r$ representing the weighting coefficients for past rewards (derived from *Equation 5*), $R_A$ as reward delivery after choice of object A, $j$ as index for past trials relative to the current $i$th trial, and $N$ as the number of past trials included in the model ($N = 10$); the term $\left( \sum_{j=1}^{N} (R_A(i-j)) \right) /N$ represents the mean reward over the last ten trials. Thus, the equation derives subjective object risk from the summed, subjectively weighted, squared deviation of reward amounts in the last ten trials from the mean reward over the last ten trials. By defining risk in this manner, we followed the common economic definition of risk as the mean squared deviation from expected outcome and in addition accounted for each animal's subjective weighting of past trials. This definition (*Equation 8*) constituted our main subjective object risk measure for behavioral and neuronal analyses.

Alternative, more complex definitions of subjective risk in our task could incorporate the weighting of past trials in the calculation of the mean reward (the subtrahend in the numerator of *Equation 8*) or incorporate both weighted reward history and weighted choice history in this calculation. We explore these possibilities in a supplementary analysis (*Figure 4—figure supplement 4*). In our main risk definition, we also assumed that the animals used a common reward weighting function for value and risk. As described in the Results, for neuronal analysis we also explored alternative past-trial weighting functions for the risk calculation (*Figure 4—figure supplement 3*). The alternative weights identified similar although slightly lower numbers of risk neurons compared to those obtained with the weights defined by *Equation 5*.

## Testing the influence of subjective object risk on choices

For out-of-sample validation, we used one half of the behavioral data within each animal to derive weighting coefficients (*Equation 5*) and subsequently used the remaining half for testing the behavioral relevance of object-risk and object-value variables. To do so, we used logistic regression to relate each animal's choices to the subjective object values and object variance-risks, according to the following equation (*Equation 9*):

$$\log\left(\frac{p_L(i)}{p_R(i)}\right) = \beta_0 + \beta_1(ObjectValueLeft - ObjectValueRight) \\ + \beta_2(ObjectRiskLeft - ObjectRiskRight) + \varepsilon \tag{9}$$

with $p_L(i)$ [or $p_R(i)$] as the probability of choosing left or right on the $i$th trial, *ObjectValueLeft* as current-trial value of the left object (derived from $OV_A^{r,c}$, *Equation 7*), *ObjectValueRight* as current-trial value of the right object (*Equation 7*), *ObjectRiskLeft* as current-trial risk of the left object (*Equation 8*), *ObjectRiskRight* as current-trial risk of the right object (*Equation 8*), $\beta_1$ to $\beta_2$ as corresponding regression coefficients, $\beta_0$ as constant, $\varepsilon$ as residual. The resulting regression coefficients are shown in *Figure 3E*. Thus, object choice was modeled as a function of relative object value and relative object risk.

We compared this model to several alternative behavioral models using Akaike Information Criterion (AIC) and Bayesian Information Criterion (BIC; *Table 1*). The alternative models included variations of the above model (described in the legend to *Table 1*), a model based on objective (true) reward probabilities and risks, a standard reinforcement learning model that updated the object-value estimate of the chosen option based on the obtained outcome (*Sutton and Barto, 1998*), a reinforcement learning model that updated object-value estimates of both chosen and unchosen option, and a modified reinforcement learning model that captured time-dependent increases in reward probability in tasks used to study matching behavior (*Huh et al., 2009*).

## Subjective risk: analysis of neuronal data

### Subjective risk for objects

Each neuronal response was tested with the following multiple regression model to identify responses related to subjective risk derived from weighted reward history (*Equation 10*):

$$y = \beta_0 + \beta_1 ObjectChoice + \beta_2 CuePosition + \beta_3 Action + \beta_4 ObjectValueA \\ + \beta_5 ObjectValueB + \beta_6 ObjectRiskA + \beta_7 ObjectRiskB + \varepsilon \tag{10}$$

with *y* as trial-by-trial neuronal impulse rate, *ObjectChoice* as current-trial object choice (0 for A, one for B), *CuePosition* as current-trial spatial cue position (0 for object A on the left, one for object A on the right), *Action* as current-trial action (0 for left, one for right), *ObjectValueA* as current-trial value of object A (*Equation 7*), *ObjectValueB* as current-trial value of object B (*Equation 7*), *ObjectRiskA* as current-trial risk of object A (*Equation 8*), *ObjectRiskB* as current-trial risk of object B (*Equation 8*), $\beta_1$ to $\beta_7$ as corresponding regression coefficients, $\beta_0$ as constant, $\varepsilon$ as residual. This equation differs from *Equation 3* as it replaced the regressors for true risk and true probability with regressors for subjective risk and subjective value. A neuronal response was classified coding object risk if it had a significant coefficient for *ObjectRiskA* or *ObjectRiskB*.

### Regression including last-trial history variables

Object risk was calculated based on reward received in previous trials. Accordingly, we used an additional regression to test whether these variables were directly encoded by DLPFC neurons, and whether object-risk responses were better explained in terms of these history variables. To test for encoding of object risk alongside explicit regressors for last-trial reward, last-trial choice, and last-trial choice × reward, we used the following regression model (*Equation 11*):

$$y = \beta_0 + \beta_1 ObjectChoice + \beta_2 CuePosition + \beta_3 Action + \beta_4 ObjectValueA \\ + \beta_5 ObjectValueB + \beta_6 ObjectRiskA + \beta_7 ObjectRiskB \\ + \beta_8 LastReward + \beta_9 LastChoice + \beta_{10} LastReward \times LastChoice + \varepsilon \tag{11}$$

To test whether inclusion of additional regressors for the past two trials affected our main results (*Figure 4—figure supplement 2*), we used the following regression model (*Equation 12*) that included regressors for rewards, choices, and their interactions on the last two trials:

$$y = \beta_0 + \beta_1 ObjectChoice + \beta_2 CuePosition + \beta_3 Action + \beta_4 ObjectValueA \\ + \beta_5 ObjectValueB + \beta_6 ObjectRiskA + \beta_7 ObjectRiskB + \beta_8 LastReward \\ + \beta_9 LastChoice + \beta_{10} LastReward \times LastChoice + \beta_{11} LastReward2 \\ + \beta_{12} LastChoice2 + \beta_{13} LastReward2 \times LastChoice2 + \varepsilon \tag{12}$$

### Step-wise regression model for testing coding of objective and subjective object risk

We used this stepwise regression as an additional analysis to test for object-risk coding and action-risk coding when these variables directly competed to explain variance in neuronal responses. Note that the objective, true probabilities used for the analysis were not the base probabilities but the trial-specific probabilities that evolved trial-by-trial from the baseline probabilities according to *Equation 1*. The following variables were included as regressors in the starting set (*Equation 13*):

$$y = \beta_0 + \beta_1 ObjectChoice + \beta_2 CuePosition + \beta_3 Action + \beta_4 ObjectValueA \\ + \beta_5 ObjectValueB + \beta_6 ObjectRiskA + \beta_7 ObjectRiskB + \beta_8 TrueProbA \\ + \beta_9 TrueProbB + \beta_{10} TrueRiskA + \beta_{11} TrueRiskB + \varepsilon \tag{13}$$

### Subjective risk for actions

To test for encoding of action risk, we used the following multiple regression model (*Equation 14*):

$$y = \beta_0 + \beta_1 ObjectChoice + \beta_2 CuePosition + \beta_3 Action + \beta_4 ActionValueL \\ + \beta_5 ActionValueR + \beta_6 ActionRiskL + \beta_7 ActionRiskR + \varepsilon \tag{14}$$

with *y* as trial-by-trial neuronal impulse rate, *ActionValueL* as current-trial value of a leftward saccade,

*ActionValueR* as current-trial value of a rightward saccade, *ActionRiskL* as current-trial risk of a leftward saccade, *ActionRiskR* as current-trial risk of a rightward saccade (all other variables as defined above). Note that subjective action values and action risks were not simply spatially referenced object values and object risks but were estimated separately, based on object reward histories and action reward histories. Specifically, regressors for action value and action risk were estimated analogously to those for object value and object risk as described in *Equation 7* and *Equation 8*, based on filter weights derived from fitting the model in *Equation 5* for action choice rather than object choice. Thus, for defining action risk, we calculated the variance of rewards that resulted from rightward and leftward saccades within the last ten trials, with coefficients from *Equation 5* (calculated for actions) determining how strongly each trial was weighted in the variance calculation. A neuronal response was classified as coding action risk if it had a significant regressor for *ActionRiskL* or *ActionRiskR*.

## Step-wise regression model for testing coding of object risk and action risk

We used this stepwise regression as an addition analysis to test for object-risk coding and action-risk coding when these variables directly competed to explain variance in neuronal responses. The following variables were included as regressors in the starting set (*Equation 15*):

$$
\begin{aligned}
y = \quad & \beta_0 + \beta_1 ObjectChoice + \beta_2 CuePosition + \beta_3 Action + \beta_4 ObjectValueA \\
& + \beta_5 ObjectValueB + \beta_6 ObjectRiskA + \beta_7 ObjectRiskB + \beta_8 ActionValueL \\
& + \beta_9 ActionValueR + \beta_{10} ActionRiskL + \beta_{11} ActionRiskR + \varepsilon
\end{aligned}
\tag{15}
$$

## Sliding window regression analysis

We used additional sliding window multiple regression analyses (using the regression model in *Equation 11*) with a 200 ms window that we moved in steps of 25 ms across each trial. To determine whether neuronal activity was significantly related to a given variable we used a bootstrap approach based on shuffled data as follows. For each neuron, we performed the sliding window regression 1000 times on trial-shuffled data and determined a false positive rate by counting the number of consecutive windows in which a regression was significant with $p<0.05$. We found that less than five per cent of neurons with trial-shuffled data showed more than six consecutive significant analysis windows. In other words, we used the shuffled data to obtain the percentage of neurons with at least one case of six consecutively significant windows. Therefore, we counted a sliding window analysis as significant if a neuron showed a significant ($p<0.05$) effect for more than six consecutive windows.

## Normalization of population activity

We subtracted from the measured impulse rate in a given task period the mean impulse rate of the control period and divided by the standard deviation of the control period (z-score normalization). Next, we distinguished neurons that showed a positive relationship to object value and those with a negative relationship, based on the sign of the regression coefficient, and sign-corrected responses with a negative relationship. Normalized activity was used for all population decoding analyses and for *Figure 4E,F* and *Figure 5E,F*.

## Normalization of regression coefficients

Standardized regression coefficients were defined as xi(si/sy), xi being the raw slope coefficient for regressor i, and si and sy the standard deviations of independent variable i and the dependent variable, respectively. These coefficients were used for *Figure 2B,C*, *Figure 3E*, *Figure 4B,C*, *Figure 5B,C*, *Figure 6C*, *Figure 7H*, *Figure 7—figure supplement 2B,C*.

## Population decoding

We used support vector machine (SVM) and nearest-neighbor (NN) classifiers to quantify the information contained in DLPFC population activity in defined task periods. This method determines how accurately our main variables object risk and action risk were encoded by groups of DLPFC neurons. The SVM classifier was trained on a set of training data to find a linear hyperplane that provides the best separation between two patterns of neuronal population activity defined by a grouping variable

(e.g. high vs. low object risk). Decoding was typically not improved by non-linear (e.g. quadratic) kernels. Both SVM and NN classification are biologically plausible in that a downstream neuron could perform similar classification by comparing the input on a given trial with a stored vector of synaptic weights. Both classifiers performed qualitatively similar, although SVM decoding was typically more accurate. We therefore focused our main results on SVM decoding.

We aggregated z-normalized trial-by-trial impulse rates of independently recorded DLPFC neurons from specific task periods into pseudo-populations. We used all recorded neurons that met inclusion criteria for a minimum trial number, without pre-selecting for risk coding, except where explicitly stated. For each decoding analysis, we created two $n$ by $m$ matrices with $n$ columns defined by the number of neurons and $m$ rows by the number of trials. We defined two matrices, one for each group for which decoding was performed (e.g. high vs. low object risk). Thus, each cell in a matrix contained the impulse rate from a single neuron on a single trial measured for a given group. Because neurons were not simultaneously recorded, we randomly matched up trials from different neurons for the same group and then repeated the decoding analysis with different random trial matching (within-group trial matching) 150 times for the SVM and 500 times for the NN. We found these numbers to produce very stable classification results. (We note that this approach likely provides a lower bound for decoding performance as it ignores potential contributions from cross-correlations between neurons; investigation of cross-correlations would require data from simultaneously recorded neurons.) We used a leave-one-out cross-validation procedure whereby a classifier was trained to learn the mapping from impulse rates to groups on all trials except one; the remaining trial was then used for testing the classifier and the procedure repeated until all trials had been tested. An alternative approach of using 80% trials as training data and testing on the remaining 20% produced highly similar results (*Pagan et al., 2013*). We only included neurons in the decoding analyses that had a minimum number of eight trials per group for which decoding was performed. 'Group' referred to a trial category for which decoding was performed, such as low risk, high risk, A chosen, B chosen, etc. The minimum defined the lower cut-off in case a recording session contained few trials that belonged to a specific group as in the case of decoding based on risk terciles within each session, separately for object A and object B.

The SVM decoding was implemented in Matlab (Version R2013b, Mathworks, Natick, MA) using the 'svmtrain' and 'svmclassify' functions with a linear kernel and the default sequential minimal optimization method for finding the separating hyperplane. We quantified decoding accuracy as the percentage of correctly classified trials, averaged over all decoding analyses for different random within-group trial matchings. To investigate how decoding accuracy depends on population size, we randomly selected a given number of neurons at each step and then determined the percentage correct. For each step (i.e. each possible population size) this procedure was repeated 10 times. We also performed decoding for randomly shuffled data (shuffled group assignment without replacement) with 5000 iterations to test whether decoding on real data differed significantly from chance. Statistical significance ($p<0.0001$) was determined by comparing vectors of percentage correct decoding accuracy between real data and randomly shuffled data using the rank sum test (*Quian Quiroga et al., 2006*). For all analyses, decoding was performed on neuronal responses taken from the same task period. We trained classifiers to distinguish high from low risk terciles (decoding based on median split produced very similar results).

## Acknowledgements

This work was supported by the Wellcome Trust (Principal Research Fellowship and Programme Grant 095495 to WS; Sir Henry Dale Fellowship 206207/Z/17/Z to FG), the European Research Council (ERC Advanced Grant 293549 to WS), and the National Institutes of Health (NIH) Caltech Conte Center (P50MH094258).

## Additional information

### Funding

| Funder | Grant reference number | Author |
|---|---|---|
| Wellcome Trust | Principal Research Fellowship | Wolfram Schultz |
| Wellcome Trust | Programme Grant 095495 | Wolfram Schultz |
| Wellcome Trust | Sir Henry Dale Fellowship 206207/Z/17/Z | Fabian Grabenhorst |
| European Research Council | Advanced Grant 293549 | Wolfram Schultz |
| National Institutes of Health | Caltech Conte Center P50MH094258 | Wolfram Schultz |

The funders had no role in study design, data collection and interpretation, or the decision to submit the work for publication.

### Author contributions

Fabian Grabenhorst, Conceptualization, Formal analysis, Investigation, Visualization, Writing—original draft, Writing—review and editing; Ken-Ichiro Tsutsui, Investigation, Methodology, Writing—review and editing; Shunsuke Kobayashi, Investigation, Writing—review and editing; Wolfram Schultz, Conceptualization, Supervision, Funding acquisition, Project administration, Writing—review and editing

### Author ORCIDs

Fabian Grabenhorst https://orcid.org/0000-0002-6455-0648
Shunsuke Kobayashi https://orcid.org/0000-0002-6868-9313
Wolfram Schultz http://orcid.org/0000-0002-8530-4518

### Ethics

Animal experimentation: All animal procedures conformed to US National Institutes of Health Guidelines and were approved by the Home Office of the United Kingdom (Home Office Project Licenses PPL 80/2416, PPL 70/8295, PPL 80/1958, PPL 80/1513). The work has been regulated, ethically reviewed and supervised by the following UK and University of Cambridge (UCam) institutions and individuals: UK Home Office, implementing the Animals (Scientific Procedures) Act 1986, Amendment Regulations 2012, and represented by the local UK Home Office Inspector; UK Animals in Science Committee; UCam Animal Welfare and Ethical Review Body (AWERB); UK National Centre for Replacement, Refinement and Reduction of Animal Experiments (NC3Rs); UCam Biomedical Service (UBS) Certificate Holder; UCam Welfare Officer; UCam Governance and Strategy Committee; UCam Named Veterinary Surgeon (NVS); UCam Named Animal Care and Welfare Officer (NACWO).

### Decision letter and Author response

Decision letter https://doi.org/10.7554/eLife.44838.032
Author response https://doi.org/10.7554/eLife.44838.033

## Additional files

### Supplementary files

• Transparent reporting form
DOI: https://doi.org/10.7554/eLife.44838.030

### Data availability

Source data files have been provided for all figures.

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
