## [Decision Letter]

Thank you for submitting your article "Primate prefrontal neurons signal economic risk derived from the statistics of recent reward experience" for consideration by *eLife*. Your article has been reviewed by three peer reviewers, one of whom is a member of our Board of Reviewing Editors, and the evaluation has been overseen by Richard Ivry as the Senior Editor. The following individual involved in review of your submission has agreed to reveal their identity: Kenway Louie (Reviewer #3).

The reviewers have discussed the reviews with one another and the Reviewing Editor has drafted this decision to help you prepare a revised submission.

Summary:

Authors of this work have examined the activity of single neurons in the monkey dorsolateral prefrontal cortex (DLPFC) during probabilistic "matching task" in which the reward probability was adjusted dynamically according to the baseline probability as well as the animal's choice history. The focus of this manuscript is to characterize the signals related to "risk", namely the uncertainty associated with the reward probability. Whereas previous studies have examined the neural signals related to "objective risk", the current study have dissected the DLPFC signals related to the experiential/subjective risk signals, and demonstrated how such risk signals co-exist with other signals previously identified in this brain area.

Essential revisions:

1) Introduction needs a better organization. Unfortunately, the term risk has been used in a few different ways, and the authors have attempted to clarify how this term is used in the present study, but the first two paragraphs of the Introduction are still somewhat confusing. The manuscript might be easier to digest, if the authors tried to clarify only the terms used in this study, and better avoid mistakenly equating the reward probability and risk (which are different) or using terms such as "variance risk", which is unnecessarily confusing. In particular, the discussion on loss and risk is not clear, and perhaps unnecessary for the paper since the focus is on variance.

2) Behavioral effects of risk on choice. Authors should provide more information about how reward risk (or variance) plays into monkey choice behavior in this task. As well documented in these type of matching law tasks, choice behavior is a product of both past rewards and past choices (as also addressed by the reward and choice kernels quantified by the authors). This arises because these kind of environments are dynamic and complex: the baited reward outcomes (probabilistic rewards that remain once armed) are constructed in order to generate matching law behavior, and thus the values of options explicitly depend on past outcomes and choices.

The question is how does risk (true or subjective) affect behavior, and is it independent of past rewards and choices? The crucial issue is whether risk (or some measure of reward variance) is capturing an aspect of behavior beyond what is captures with past rewards and choices alone. In the analyses in Figure 3, the authors decompose task-relevant information into object value and subjective object risk, and show that both have an effect on monkey choice. However, object value is the weighted sum of reward history (with the weights determined by regression on overall monkey choices), meaning that the effect of past choices is explicitly not captured in the value variable. The key question is whether subjective risk is just capturing the effect of past choices, and I think the authors need to find a way of quantifying the relative influence of risk – above and beyond past choices – on behavior. One way to be to do a formal comparison between the logistic regression based on past rewards and choice and a model using value and risk, or alternatively a full model with rewards, choices, AND risk in the same model. [The authors may have included this kind of analysis (as seems to be the case in the model comparison noted in the Materials and methods, subsection “Testing the influence of subjective object risk on choices”), but the description in the main text only refers to value as a function of past rewards – please correct me if I am mistaken.]

Understanding of risk, as defined, affects choice behavior is important two reasons. At the behavioral level, it is not yet clear to me how reward risk – objective or subjective – is related to monkey choice behavior in this experiment. One possibility, as detailed above, is that risk is merely capturing the effect of past monkey choices. Alternatively, monkeys may have a preference/aversion for risk itself. The latter point is what I believe the authors are getting at in their analyses (subsection “Subjective risk: definition and behavior”, last paragraph, Figure 3D), but as discussed above this is done by examining the influence of risk on choices for various object value differences, with object values determined solely by reward kernel weighting – this ignores the influence of past choices that may well govern the effect of risk in this task. At the neural level, the authors electrophysiological results show a robust coding for risk in DLPFC neurons, and it is important to distinguish whether these neurons are coding for (objective or subjective) risk itself, or simply some aspect of choice-related strategy that correlates with risk.

3) Methods to quantify subjective needs to be justified better or improved. The authors have demonstrated that the animals incorporated their choice history to determine their values (Equation 5). Given this, it seems difficult to justify that choice history is not incorporated in the estimation of risk (Equation 6 and 7).

First, Equation 6 (that estimates value) does so in a classical RL way by assigning higher values to options that have been recently rewarded (more often). However, in this task the true value of an option is considerably lower after having been recently chosen (Equation 1 says the reward probability is lowest after a choice and then grows with each trial the option is not chosen) and animals seem to understand this as shown by their *β_c_* weights. Shouldn't the subjective value estimate be based on both past reward and past choice since both influence the animal's choice? The same point applies to Equation 7. If animals understand the task, they should know that the option's risk on the current trial is influenced by its choice history in addition to its reward history.

Second, Equation 7 attempts to estimate variance. It does not seem to be a true variance measure (i.e. mean squared deviation of quantities from the mean of their distribution) because it is based on the deviation from the sum of those quantities (*OV_A_*, from Equation 6) not their mean.

Beyond that, an estimate of variance should involve the variance of past outcomes, but Equation 7 is also strongly influenced by the variance of the β weights *β_r_* and the variance of choice history, and treats reward and non-reward asymmetrically. For instance, if the animal chose option A 10 consecutive times and always received reward, then the reward history has zero variance but *var_A_* will still be high because of the variance in the *β_r_*. Also, if the animal chose A 10 consecutive times and always received no reward, then *var_A_* will be zero, even though it is symmetrical to the previous case and should have the same variance. Finally, if the animal chose B 5 times and then chose A 5 times and always received reward, then *var_A_* will be higher than if the animal chose A all 10 times and always received reward, even though the variance of the reward histories from choices of A are the same (5/5 vs. 10/10 rewards), because of the variance in choice history.

The authors must either 1) use another modelling approach that considers the structure of the "dual assignment with hold" task or 2) substantially revise to make it clear to the reader what the limitations are of the current approach and why they chose to utilize it.

4) Relationship between neural signals related to objective and subjective risk. One important aspect, however, that is not addressed is the relationship between true risk (i.e. Equation 2) and subjective risk (i.e. Equation 7). How correlated are these measures? At the behavioral level, does subjective risk do a better job at explaining choices compared to true risk?

At the neural level, the major conclusions of the paper center around the DLPFC coding of subjective object and action risk, but the paper does not clearly show that subjective risk better explains neural responses than true objective risk. For example, based on the stated results, 102 of 205 neurons (subsection “Neuronal coding of objective risk”, second paragraph) significantly responded to true object risk and 96 of 205 neurons (subsection “Neuronal coding of subjective risk associated with choice objects”, first paragraph) coded for subjective object risk. Is there a way for the authors to statistically distinguish whether DLPFC is encoding subjective rather than objective risk? If the different risk measures are uncorrelated (or only mildly correlated), this should be testable; if they are strongly correlated, I am not sure that the neural analyses centered on subjective risk (rather than true risk) are the right approach.

Showing that the authors' subjective risk estimate better captures choice and/or neural data is important because the specific quantification of subjective risk is not well known or clearly justified. Aside from the issue of the potential relationship between past choice effects and risk (see point 1 above), the weighting of past rewards in the variance calculation feels a bit arbitrary (not the reward kernel itself, which is well known in quantifying choice, but its use in estimating variance). In addition to showing that such a measure better captures behavior/neural responses, it would help if the authors could provide a more formal justification for their form of subjective risk.

5) Potential problems and weaknesses of decoding analysis. The decoding analysis needs to be strengthened, because currently this analysis does not attempt to distinguish between risk signals and other potentially covarying signals (as was done in the regression analysis). This can be accomplished for example, by balancing the trials with low and high levels of risk, in terms of other potentially confounding variables (c.f. Massi et al., 2018).

The% of neurons plots common to neuroeconomics studies (and SVM-style decoding that only shows what info is in a pool of neurons and not how good individual neurons are at "encoding") may not be well suited for dlPFC. dlPFC is a complex spatial- and object- selective, attention, and motivation/reward related area and accordingly multiplexes many signals. It is simply very hard to tell what is going on from the current figures on a cell by cell basis. I want to see if the coding strategies (e.g. value, risk) of the single neurons in dlPFC are consistent across the task epochs in this task to get a better sense of what dlPFC may be doing, and to get a better understanding of the relationship of those value-related variables with spatial and object preferences of single neurons.

6) Description of models and equations need to be improved.

6a) The authors introduce two equations (Equations 2 and 7) to define objective and subjective risks, respectively, which are central to this study. However, these two equations can be simplified. For example, it would be much easier to understand this if the Equation 2 is replaced or supplemented by a much more common expression, p(1-p), for a Bernoulli distribution. Including the reward magnitude in Equation 2 doesn't seem necessary, and this causes confusion. In addition, in Equation 7, the coefficient β should be outside the square of the difference between reward and OV. If not, this this needs to be justified/explained better.

6b) The authors have used Equation 2 in their step-wise regression analysis. However, this seems problematic, because this seems to violate the full-rank assumption, given that ValueA+ValueB = ValueL+ValueR and RiskA+RiskB=RiskL+RiskR? Similarly, is it possible to have both TrueProbA and TrueProbB in Equation 3, given that they sum up to 1, i.e., TrueProbB = (1-TrueProbA)?

6c) The exact formulation of Equation 8 is a little unclear. The text states that "To account for known choice biases in matching tasks (Lau and Glimcher, 2005), we added to the object-value terms the weighted choice-history using weights derived from Equation 5." Can the authors state explicitly how ObjectValue was calculated?

7) Novelty of the task. The authors claim that the animals are not cued and must derive a measure of risk (that would then influence their choices). This important because the authors claim that previous studies used explicit cues during learning while they do not in this study, and claim this is a key advance. While it is true, the cited papers did not utilize trial-by-trial information to look at how subjective value- and risk- are updated on a trial-by-trial basis, their approaches in many other ways seemed pretty similar to this study. Particularly, in this study, the key moment is when the reward probabilities associated with the two options change and the animal must figure out the new probabilities by experiencing two types of external cues: choice options and feedback. Broadly, the idea that this is the first paper to look at risk estimates that are independent of cueing is factually wrong and requires revision.

Furthermore, even if it was the case, the impact of this is unclear from the current manuscript. Perhaps the authors are most excited about pre-option-presentation object-risk signals in the context of behavioral control? If so, how would this control take place? Is this an arousal / "let's get ready" signal? Or would this signal serve to bias choice to the risky object in a spatial manner (consistent with previous work on dlPFC)? Or is this risk signal to influence SV derivations elsewhere in the brain?

Specifically, are there reaction time correlates of this early "object risk signal" with action after options are on?

8) Problem with non-stationarity. The authors should take into account the fact that activity of prefrontal cortex is often non-stationary and is likely to be correlated serially (autocorrelation) across successive trials. This diminishes the effective degree of freedom, and can inflate the estimate of neurons encoding the signals that are related to events in multiple trials (equivalent to low-pass filtering). The authors should refer to a recent paper in *eLife* ("Striatal action-value neurons reconsidered).

9) Dynamics of DLPFC coding. The authors show that subpopulations of neurons carry multiple signals that could integrate various aspects of reward and choice information (Figure 7). Two additional analyses are important to include. First, is the percentage of neurons showing coding of two variables (last reward x choice and object risk, object value and object risk, etc.) different than that expected given the probabilities of neurons representing either variable?

Second, for those neurons that carry multiple signals, is the information coded in a consistent manner? For example, do the neurons that represent both value and risk information (Figure C, D) both modulated in the same direction by value and risk? Figure 7 plots the timeline of explained variance, but not the actual direction of modulation. One would expect that, for example in the case of value and risk, that since the behavioral data suggests that choice is driven by increases in value and risk, that neurons integrating that information would represent both in the same manner. A similar argument could be made for risk and choice. A regression weight by regression weight plot, similar to that used for the angle analyses elsewhere in the paper, would be helpful in understanding how this information is integrated across different variable pairs.

[Editors' note: further revisions were requested prior to acceptance, as described below.]

Thank you for resubmitting your work entitled "Primate prefrontal neurons signal economic risk derived from the statistics of recent reward experience" for further consideration at *eLife*. Your revised article has been favorably evaluated by Richard Ivry as the Senior Editor, a Reviewing Editor, and two reviewers.

The manuscript has been improved but there are some remaining issues that need to be addressed before acceptance, as outlined below:

The authors have largely addressed the concern re: the influence of past choices on subjective value, and Table 1 shows that a model with value (from reward and choice history) and risk performs best. However, the present manuscript is still confusing in that it includes two different measure of subjective object value (Equation 6 and Equation 7).

First, it is confusing that the two equations both use the same term *OV_A_* for the value measure; since they are different definitions, they should have different names. Second, it is entirely not clear which measure is used in which analyses. According to the Materials and methods, Equation 6 (value from rewards alone) is used for the calculation of subjective risk (Equation 8), and Equation 7 (value from rewards and choices) is used for behavioral analyses and neural analyses involving value. So were all neural subjective risk analyses performed only with the measure derived from form Equation 6 *OV_A_*? This seems odd, given that the results of the model comparison suggest that value is a function of reward and choice, and the authors use the reward/choice definition of value for neural analyses – shouldn't the subjective risk measure use deviance from *OV_A_* determined from reward and choice as well?

The paper would read more clearly, and be more conceptually unified, if they use a single measure for *OV_A_* derived from reward and past choices (Equation 8). Note that this is different than the "risk from choice variance" addressed by the authors in their response letter – it is simply risk as variance of rewards from subjective value (calculated from reward and choice).

The description in the Materials and methods (subsection “Final calculation of subjective risk”) is incorrect given the revised Equation 8, as it implies that β weights are applied to rewards rather than to the squared deviations of rewards from *OV_A_* (as described by the revised equation). For example, "βjrRAβ" term is not in the equation, and "the summed, squared deviation of subjectively weighted reward amounts from the mean weighted value of past rewards)" does not match Equation 8.

---

## [Author Response]

Essential revisions:1) Introduction needs a better organization. Unfortunately, the term risk has been used in a few different ways, and the authors have attempted to clarify how this term is used in the present study, but the first two paragraphs of the Introduction are still somewhat confusing. The manuscript might be easier to digest, if the authors tried to clarify only the terms used in this study, and better avoid mistakenly equating the reward probability and risk (which are different) or using terms such as "variance risk", which is unnecessarily confusing. In particular, the discussion on loss and risk is not clear, and perhaps unnecessary for the paper since the focus is on variance.

Thank you for pointing out the need for better organization. We have revised the Introduction accordingly and now focus on the terms used in the present study.

“Rewards vary intrinsically. The variation can be characterized by a probability distribution over reward magnitudes. […] Thus, among the different definitions of economic risk, variance constitutes the most basic form, and this study will consider only variance as economic risk.”

2) Behavioral effects of risk on choice. Authors should provide more information about how reward risk (or variance) plays into monkey choice behavior in this task. As well documented in these type of matching law tasks, choice behavior is a product of both past rewards and past choices (as also addressed by the reward and choice kernels quantified by the authors). This arises because these kind of environments are dynamic and complex: the baited reward outcomes (probabilistic rewards that remain once armed) are constructed in order to generate matching law behavior, and thus the values of options explicitly depend on past outcomes and choices.The question is how does risk (true or subjective) affect behavior, and is it independent of past rewards and choices? The crucial issue is whether risk (or some measure of reward variance) is capturing an aspect of behavior beyond what is captures with past rewards and choices alone. In the analyses in Figure 3, the authors decompose task-relevant information into object value and subjective object risk, and show that both have an effect on monkey choice. However, object value is the weighted sum of reward history (with the weights determined by regression on overall monkey choices), meaning that the effect of past choices is explicitly not captured in the value variable. The key question is whether subjective risk is just capturing the effect of past choices, and I think the authors need to find a way of quantifying the relative influence of risk – above and beyond past choices – on behavior. One way to be to do a formal comparison between the logistic regression based on past rewards and choice and a model using value and risk, or alternatively a full model with rewards, choices, AND risk in the same model. [The authors may have included this kind of analysis (as seems to be the case in the model comparison noted in the Materials and methods, subsection “Testing the influence of subjective object risk on choices”), but the description in the main text only refers to value as a function of past rewards – please correct me if I am mistaken.]Understanding of risk, as defined, affects choice behavior is important two reasons. At the behavioral level, it is not yet clear to me how reward risk – objective or subjective – is related to monkey choice behavior in this experiment. One possibility, as detailed above, is that risk is merely capturing the effect of past monkey choices. Alternatively, monkeys may have a preference/aversion for risk itself. The latter point is what I believe the authors are getting at in their analyses (subsection “Subjective risk: definition and behavior”, last paragraph, Figure 3D), but as discussed above this is done by examining the influence of risk on choices for various object value differences, with object values determined solely by reward kernel weighting – this ignores the influence of past choices that may well govern the effect of risk in this task. At the neural level, the authors electrophysiological results show a robust coding for risk in DLPFC neurons, and it is important to distinguish whether these neurons are coding for (objective or subjective) risk itself, or simply some aspect of choice-related strategy that correlates with risk.

We have included a formal comparison of different models of the animals’ behavioral choices. Specifically, we systematically compared models that included different forms of subjective values, with values based on weighted reward history, weighted choice history or both weighted reward and weighted choice history. We compared the effect of adding our main risk measure as a separate regressor to these different forms of value. We also tested a model based on true, objective reward probabilities and risk. Finally, we tested three versions of reinforcement learning models: (i) a standard Rescorla-Wagner model that updated the value of the chosen option following outcomes, (ii) an adaptation of this model that incorporated time-dependent effects related to choice history, which has been proposed as a suitable model of matching behavior (Huh et al., 2009), and (iii) a variant of the Rescorla-Wagner model that updated both the value of the chosen and unchosen option. The results are shown in a new table (Table 1).

In both animals, the model comparisons favored a model that included subjective value and subjective risk regressors, with subjective value based on both weighted reward and choice history. This result confirms that our main measure of subjective risk was behaviorally meaningful and explained variation in choices that is independent of reward history and choice history.

To further illustrate this point, we have performed a new logistic regression of choices on value and risk in a subset of trials that minimized the value difference between options. For these trials, value difference did not explain variation in choices (as expected by design of this test) whereas the effect of risk remained significant. Thus, the effect of risk on choices was not explained by value difference. The result of this analysis is shown in Figure 3E, inset and described in the last paragraph of the Results subsection “Subjective risk: definition and behavior”.

3) Methods to quantify subjective needs to be justified better or improved. The authors have demonstrated that the animals incorporated their choice history to determine their values (Equation 5). Given this, it seems difficult to justify that choice history is not incorporated in the estimation of risk (Equations 6 and 7).First, Equation 6 (that estimates value) does so in a classical RL way by assigning higher values to options that have been recently rewarded (more often). However, in this task the true value of an option is considerably lower after having been recently chosen (Equation 1 says the reward probability is lowest after a choice and then grows with each trial the option is not chosen) and animals seem to understand this as shown by their β_c_ weights. Shouldn't the subjective value estimate be based on both past reward and past choice since both influence the animal's choice? The same point applies to Equation 7. If animals understand the task, they should know that the option's risk on the current trial is influenced by its choice history in addition to its reward history.Second, Equation 7 attempts to estimate variance. It does not seem to be a true variance measure (i.e. mean squared deviation of quantities from the mean of their distribution) because it is based on the deviation from the sum of those quantities (OV_A_, from Equation 6) not their mean.Beyond that, an estimate of variance should involve the variance of past outcomes, but Equation 7 is also strongly influenced by the variance of the β weights β_r_ and the variance of choice history, and treats reward and non-reward asymmetrically. For instance, if the animal chose option A 10 consecutive times and always received reward, then the reward history has zero variance but var_A_ will still be high because of the variance in the β_r_. Also, if the animal chose A 10 consecutive times and always received no reward, then var_A_ will be zero, even though it is symmetrical to the previous case and should have the same variance. Finally, if the animal chose B 5 times and then chose A 5 times and always received reward, then var_A_ will be higher than if the animal chose A all 10 times and always received reward, even though the variance of the reward histories from choices of A are the same (5/5 vs. 10/10 rewards), because of the variance in choice history.The authors must either 1) use another modelling approach that considers the structure of the "dual assignment with hold" task or 2) substantially revise to make it clear to the reader what the limitations are of the current approach and why they chose to utilize it.

We thank the reviewers for raising these issues and pointing out the need for clearer and better-justified definitions. In responding to the points raised we have revised our value definition to incorporate choice history and have recalculated all our main analyses accordingly and updated all relevant figures. We now also explicitly discuss the assumptions and limitations of our approach to defining subjective risk. Below we explain these changes in more detail.

1,1) Should value include choice history? We agree that it is important to use a comprehensive definition of value for behavioral and neuronal analyses. Accordingly, we have now modelled value in direct correspondence to our logistic regression model have now incorporated this component into one scalar value measure. Accordingly, we recalculated all our models with this revised, more comprehensive value definition, which resulted in small changes in the numbers of identified neurons. We have updated all the numbers of identified neurons and all relevant figures to reflect these changes. The new value definition is described in Results, and in Materials and methods, section “Calculation of weighted, subjective value.”

Results: “As our aim was to study the risk associated with specific objects, we estimated object value by the mean of subjectively weighted reward history over the past ten trials (Figure 3C, dashed blue curve, Equation 6); this object value definition provided the basis for estimating subjective risk as described next. […] (We consider distinctions between reward and choice history and their influence and risk in the Discussion).”

1,2) Should risk include choice history? One of the main aims of our study was to extend the well-established notion of the risk of choice options by introducing a risk measure derived from an animal’s recent experiences, rather than from pre-trained explicit risk cues. Accordingly, in order to facilitate comparisons with previous behavioral and neurophysiological risk studies, we restricted our definition of risk to the variance of past reward outcomes; we did not extend the risk measure to include the variance of past choices irrespective of rewards as such “risk from choice variance” does not have a correspondence in the economic or neuroeconomic risk literature. Our results suggest that defining risk based on the variance of past rewards provided a reasonable account of behavioral and neuronal data: we show that our main subjective risk measure has a distinct influence on choices, is encoded by a substantial number of neurons, and that this risk measure seems to provide a better explanation of many neuronal responses compared to an alternative, objective risk measure. To better explain our motivation for defining risk based on reward history, we have included the following additional text in Discussion:

“Our definition of subjective risk has some limitations. To facilitate comparisons with previous studies, we restricted our definition to the variance of past reward outcomes; we did not extend the risk measure to the variance of past choices, which would not have a clear correspondence in the economic or neuroeconomic literature. […] An extension of this approach could introduce calculation of reward statistics over flexible time windows, possibly depending on the number of times an option was recently chosen (similar to an adaptive learning rate in reinforcement learning models).”

2) Should risk involve subtraction of value calculated from sum or mean? Thank you for pointing this out. For the results presented previously in the manuscript we did follow the general definition of risk and subtracted the mean (rather than the sum) although we used the sum for value definition. (Note that for behavioral and neuronal analyses that test the effect of value, using the sum or mean of the weighted reward history would yield identical results as the number of trials over which the mean is calculated is constant (N = 10 in our case)). For the variance calculation it is of course critical that deviations of single trials are referenced to the mean rather than the sum. We have now corrected the equation in the Materials and methods (Equation 6).

3) Influence of weights and choice history on variance. We have rewritten the risk equation (now Equation 8) to reflect correctly how we calculated risk: the weight vector was applied to the vector of squared deviations from the mean. With this definition, deviations calculated for more recent trials are given a larger weight in the variance calculation. This approach is similar to the well-established definition of value based on weighted reward history and it is consistent with the notion that the animals base their choice more strongly on reward outcomes of recent trials compared to more remote trials.

With this definition, variance would not be artificially inflated by the weight vector, and reward and non-reward are not treated asymmetrically. For example, if the animal chose option A 10 consecutive times and always received reward, then reward history has zero variance and multiplication with the weight vector still results in zero variance. The same would be true for the case of 10 consecutively non-rewarded trials.

The reviewer notes correctly that “if the animal chose B 5 times and then chose A 5 times and always received reward, then *var_A_* will be higher than if the animal chose A all 10 times and always received reward”. This is a consequence of the time window (here: 10 trials) used for the variance calculation that also affects the calculation of value in a similar manner. We chose to follow this common approach of using a fixed temporal window over which reward statistics are calculated because of its generality and simplicity, and in order to link our study with these established approaches. An extension of this approach would be to introduce a flexible temporal window that calculates reward statistics over varying time windows, possibly depending on the number of times an option was recently chosen (similar to an adaptive learning rate in Reinforcement learning models).

To acknowledge the assumptions and limitations of our risk definition we have included the following new text in the Discussion section:

“We followed the common approach of calculating reward statistics over a fixed temporal window because of its generality and simplicity, and to link our data to previous studies. An extension of this approach could introduce calculation of reward statistics over flexible time windows, possibly depending on the number of times an option was recently chosen (similar to an adaptive learning rate in reinforcement learning models).”

4) Relationship between neural signals related to objective and subjective risk. One important aspect, however, that is not addressed is the relationship between true risk (i.e. Equation 2) and subjective risk (i.e. Equation 7). How correlated are these measures? At the behavioral level, does subjective risk do a better job at explaining choices compared to true risk?At the neural level, the major conclusions of the paper center around the DLPFC coding of subjective object and action risk, but the paper does not clearly show that subjective risk better explains neural responses than true objective risk. For example, based on the stated results, 102 of 205 neurons (subsection “Neuronal coding of objective risk”, second paragraph) significantly responded to true object risk and 96 of 205 neurons (subsection “Neuronal coding of subjective risk associated with choice objects”, first paragraph) coded for subjective object risk. Is there a way for the authors to statistically distinguish whether DLPFC is encoding subjective rather than objective risk? If the different risk measures are uncorrelated (or only mildly correlated), this should be testable; if they are strongly correlated, I am not sure that the neural analyses centered on subjective risk (rather than true risk) are the right approach.Showing that the authors' subjective risk estimate better captures choice and/or neural data is important because the specific quantification of subjective risk is not well known or clearly justified. Aside from the issue of the potential relationship between past choice effects and risk (see point 1 above), the weighting of past rewards in the variance calculation feels a bit arbitrary (not the reward kernel itself, which is well known in quantifying choice, but its use in estimating variance). In addition to showing that such a measure better captures behavior/neural responses, it would help if the authors could provide a more formal justification for their form of subjective risk.

The objective and subjective risk measures only showed a moderate correlation: mean shared variance was R^2^ = 0.111 ± 0.004 (mean ± s.e.m. across sessions). To establish their relevance for behavioral choices, we have included a formal model comparison, summarized in Table 1, which favored a model based on subjective value and subjective risk over a model based on objective value and objective risk. We also examined in direct comparisons whether neuronal responses were better explained by objective or subjective risk. These analyses are described in Results and in Figure 4—figure supplement 3B and C:

“A direct comparison of objective and subjective risk showed that neuronal activity tended to be better explained by subjective risk. […] When both risk measures were included in a stepwise regression model (Equation 13), and thus competed to explain variance in neuronal activity, we identified more neurons related to subjective risk than to objective risk (107 compared to 83 neurons, Figure 4—figure supplement 3C), of which 101 neurons were exclusively related to subjective risk but not objective risk (shared variance between the two risk measures across sessions: R^2^ = 0.111 ± 0.004, mean ± s.e.m.).”

5) Potential problems and weaknesses of decoding analysis. The decoding analysis needs to be strengthened, because currently this analysis does not attempt to distinguish between risk signals and other potentially covarying signals (as was done in the regression analysis). This can be accomplished for example, by balancing the trials with low and high levels of risk, in terms of other potentially confounding variables (c.f. Massi et al., 2018).The% of neurons plots common to neuroeconomics studies (and SVM-style decoding that only shows what info is in a pool of neurons and not how good individual neurons are at "encoding") may not be well suited for dlPFC. dlPFC is a complex spatial- and object- selective, attention, and motivation/reward related area and accordingly multiplexes many signals. It is simply very hard to tell what is going on from the current figures on a cell by cell basis. I want to see if the coding strategies (e.g. value, risk) of the single neurons in dlPFC are consistent across the task epochs in this task to get a better sense of what dlPFC may be doing, and to get a better understanding of the relationship of those value-related variables with spatial and object preferences of single neurons.

We have performed additional decoding analyses in which we balanced risk levels with respect to other task-related variables. The results of these analyses are shown in Figure 6E and confirm significant decoding of risk levels and are described in Results:

“Decoding of risk from neuronal responses remained significantly above chance in control analyses in which we held constant the value of other task-related variables including object choice, action and cue position (Figure 6E).”

We also clarify that we used these decoding analyses to examine the extent to which a biologically realistic decoder, such as a downstream neurons could read out risk levels from neuronal population responses, rather than to provide an alternative to the single-neuron regression analysis (subsection “Population decoding of object risk and action risk”). Such a downstream neuron decoding risk from its inputs would of course need to perform the decoding on naturally occurring, “unbalanced” data.

We thank the reviewer(s) for raising the interesting issue of the complexity of neuronal responses in DLPFC, and their relationships to spatial variables. We have now examined this issue in more detail in our data set and have included these analyses in Figure 7I, Figure 7—figure supplement 1, and described in the Results and in the Discussion.

Results: “The percentages of neurons coding specific pairs of variables was not significantly different than expected given the probabilities of neurons coding each individual variable (history and risk: χ2 = 1.58, P = 0.2094, value and risk: χ2 = 3.54, P = 0.0599, choice and risk: χ2 = 0.845, P = 0.358). We also tested for relationships in the coding scheme (measured by signed regression coefficients) among neurons with joint risk and choice coding or joint risk and value coding. […] In addition to the risk-related dynamic coding transitions described above, activity in some DLPFC neurons transitioned from coding risk to coding of spatial variables such as cue position or action choice (Figure 7—figure supplement 1).”

Discussion: “Notably, while some DLPFC neurons jointly coded risk with value or choice in a common coding scheme (indicated by regression coefficients of equal sign), this was not the rule across all neurons with joint coding (Figure 7H). […] An implication for the present study might be that risk signals in DLPFC can support multiple cognitive processes in addition to decision-making, as also suggested by the observed relationship between risk and reaction times (Figure 3—figure supplement 1).”

6) Description of models and equations need to be improved.6a) The authors introduce two equations (Equations 2 and 7) to define objective and subjective risks, respectively, which are central to this study. However, these two equations can be simplified. For example, it would be much easier to understand this if the Equation 2 is replaced or supplemented by a much more common expression, p(1-p), for a Bernoulli distribution. Including the reward magnitude in Equation 2 doesn't seem necessary, and this causes confusion. In addition, in Equation 7, the coefficient β should be outside the square of the difference between reward and OV. If not, this this needs to be justified/explained better.

Thank you for pointing out the misplaced β in Equation 7 which we have corrected (now Equation 8). We prefer to keep the reward magnitude term in Equation 2 as the definition of variance in this notation is consistent with previous neuroscientific studies of risk and readily generalizes to situations in which different magnitudes are used. We have included a statement that explains this below the equation: “In our task, reward magnitude on rewarded trials was held constant at 0.7 ml; the definition generalizes to situations with different magnitudes.”

6b) The authors have used Equation 2 in their step-wise regression analysis. However, this seems problematic, because this seems to violate the full-rank assumption, given that ValueA+ValueB = ValueL+ValueR and RiskA+RiskB=RiskL+RiskR? Similarly, is it possible to have both TrueProbA and TrueProbB in Equation 3, given that they sum up to 1, i.e., TrueProbB = (1-TrueProbA)?

Note that subjective action values and action risks were not simply spatially referenced object values and object risks but were estimated separately, based on object reward histories and action reward histories. Accordingly, the stepwise regression approach was not invalidated by the joint inclusion of these regressors in the starting set. Moreover, the true probabilities used for the analysis were not the base probabilities but the trial-specific probabilities that evolved trial-by-trial from the baseline probabilities according to Equation 1. We have clarified these points in the Materials and methods section: “Note that subjective action values and action risks were not simply spatially referenced object values and object risks but were estimated separately, based on object reward histories and action reward histories.”

6c) The exact formulation of Equation 8 is a little unclear. The text states that "To account for known choice biases in matching tasks (Lau and Glimcher, 2005), we added to the object-value terms the weighted choice-history using weights derived from Equation 5." Can the authors state explicitly how ObjectValue was calculated?

We have now clarified this section by introducing the new Equation 7 and related new text.

7) Novelty of the task. The authors claim that the animals are not cued and must derive a measure of risk (that would then influence their choices). This important because the authors claim that previous studies used explicit cues during learning while they do not in this study, and claim this is a key advance. While it is true, the cited papers did not utilize trial-by-trial information to look at how subjective value- and risk- are updated on a trial-by-trial basis, their approaches in many other ways seemed pretty similar to this study. Particularly, in this study, the key moment is when the reward probabilities associated with the two options change and the animal must figure out the new probabilities by experiencing two types of external cues: choice options and feedback. Broadly, the idea that this is the first paper to look at risk estimates that are independent of cueing is factually wrong and requires revision.Furthermore, even if it was the case, the impact of this is unclear from the current manuscript. Perhaps the authors are most excited about pre-option-presentation object-risk signals in the context of behavioral control? If so, how would this control take place? Is this an arousal / "let's get ready" signal? Or would this signal serve to bias choice to the risky object in a spatial manner (consistent with previous work on dlPFC)? Or is this risk signal to influence SV derivations elsewhere in the brain?Specifically, are there reaction time correlates of this early "object risk signal" with action after options are on?

Novelty of task: We have toned down the aspect of cue-independence throughout. In the Introduction we added the following sentence” “Similar to previous studies (cited above), experienced rewards following choices for specific objects or actions constituted external cues for risk estimation.” To acknowledge that rewards and choice options of course constituted critical cues for risk estimation, we also removed the emphasis “without requiring explicit, risk-informative cues.” from the last sentence of the Introduction and we removed “in the absence of explicit risk information” from the first sentence of the Discussion. In the first paragraph of the Discussion, when referring to explicit cues, we have added the following: “(such as pre-trained risk-associated bar stimuli or fractals)”. We have also revised the Abstract accordingly.

Influences on behavior: The Results section ‘Dynamic integration of risk with reward history, value and choice in single neurons’ provides evidence for how DLPFC neurons may integrate risk with choice and value, and we have included new analyses to show that there is also integration with spatially referenced variables in Figure 7—figure supplement 1: “In addition to the risk-related dynamic coding transitions described above, activity in some DLPFC neurons transitioned from coding risk to coding of spatial variables such as cue position or action choice (Figure 7—figure supplement 1).” Moreover, the Discussion covers this topic in several places (sixth and seventh paragraphs, and dedicated Discussion paragraphs covering DLPFC’s contribution to risk and decision-making processes). Taken together, we acknowledge that risk signals in DLPFC may support several functions, in addition to influencing choices, either through local processing or connections to other brain structures.

Reaction time correlates We have performed a new analysis in which we regress saccadic reaction times on value and risk variables and other factors. These results are shown in Figure 3—figure supplement 1 and mentioned in the Results (subsection “Neuronal coding of subjective risk associated with actions”) and Discussion (sixth paragraph).

8) Problem with non-stationarity. The authors should take into account the fact that activity of prefrontal cortex is often non-stationary and is likely to be correlated serially (autocorrelation) across successive trials. This diminishes the effective degree of freedom, and can inflate the estimate of neurons encoding the signals that are related to events in multiple trials (equivalent to low-pass filtering). The authors should refer to a recent paper in eLife ("Striatal action-value neurons reconsidered).

We have addressed this issue with control analyses as described below. We note that in task such as the present one, probabilities and associated risk change and reset frequently even within trial blocks; accordingly, related neuronal signals tracking value or risk should be quite distinct from any potential non-stationary activity due to noise, drift or unknown sources. We explored whether potential non-stationarity in neuronal activity could have inflated estimates of risk-coding signals, as risk evolved over trials. We performed two control analyses and include these results as follows:

“Finally, we examined effects of potential non-stationarity of neuronal activity (Elber-Dorozko and Loewenstein, 2018), by including a first order autoregressive term in Equation 10. […] This analysis identified 56 neurons with activity related to risk (note that the control period itself was excluded from this analysis; our original analysis without the control period yields 81 risk neurons).”

“Controlling for non-stationarity of neuronal responses, we identified 83 action-risk neurons when including a first-order autoregressive term and 56 neurons when subtracting neuronal activity at trial start.”

9) Dynamics of DLPFC coding. The authors show that subpopulations of neurons carry multiple signals that could integrate various aspects of reward and choice information (Figure 7). Two additional analyses are important to include. First, is the percentage of neurons showing coding of two variables (last reward x choice and object risk, object value and object risk, etc.) different than that expected given the probabilities of neurons representing either variable?Second, for those neurons that carry multiple signals, is the information coded in a consistent manner? For example, do the neurons that represent both value and risk information (Figure C, D) both modulated in the same direction by value and risk? Figure 7 plots the timeline of explained variance, but not the actual direction of modulation. One would expect that, for example in the case of value and risk, that since the behavioral data suggests that choice is driven by increases in value and risk, that neurons integrating that information would represent both in the same manner. A similar argument could be made for risk and choice. A regression weight by regression weight plot, similar to that used for the angle analyses elsewhere in the paper, would be helpful in understanding how this information is integrated across different variable pairs.

Thank you for suggesting these new analyses which we have now included in Results, Figure 7H, and Discussion:

Results: “The percentages of neurons coding specific pairs of variables was not significantly different than expected given the probabilities of neurons coding each individual variable (history and risk: χ2 = 1.58, P = 0.2094, value and risk: χ2 = 3.54, P = 0.0599, choice and risk: χ2 = 0.845, P = 0.358). We also tested for relationships in the coding scheme (measured by signed regression coefficients) among neurons with joint risk and choice coding or joint risk and value coding. […] This suggested that while some neurons used corresponding coding schemes for these variables (risk and choice, risk and value) other neurons used opposing coding schemes (see Discussion for further interpretation).”

Discussion: “Notably, while some DLPFC neurons jointly coded risk with value or choice in a common coding scheme (indicated by regression coefficients of equal sign), this was not the rule across all neurons with joint coding (Figure 7H). This result and the observed high degree of joint coding, with most DLPFC dynamically coding several task-related variables (Figure 7I), matches well with previous reports that neurons in DLPFC show heterogeneous coding and mixed selectivity (Rigotti et al., 2013; Wallis and Kennerley, 2010).”

[Editors' note: further revisions were requested prior to acceptance, as described below.]

The manuscript has been improved but there are some remaining issues that need to be addressed before acceptance, as outlined below:The authors have largely addressed the concern re: the influence of past choices on subjective value, and Table 1 shows that a model with value (from reward and choice history) and risk performs best. However, the present manuscript is still confusing in that it includes two different measure of subjective object value (Equation 6 and Equation 7).First, it is confusing that the two equations both use the same term OV_A_ for the value measure; since they are different definitions, they should have different names. Second, it is entirely not clear which measure is used in which analyses. According to the Materials and methods, Equation 6 (value from rewards alone) is used for the calculation of subjective risk (Equation 8), and Equation 7 (value from rewards and choices) is used for behavioral analyses and neural analyses involving value. So were all neural subjective risk analyses performed only with the measure derived from form Equation 6 OV_A_? This seems odd, given that the results of the model comparison suggest that value is a function of reward and choice, and the authors use the reward/choice definition of value for neural analyses – shouldn't the subjective risk measure use deviance from OV_A_ determined from reward and choice as well?The paper would read more clearly, and be more conceptually unified, if they use a single measure for OV_A_ derived from reward and past choices (Equation 8). Note that this is different than the "risk from choice variance" addressed by the authors in their response letter – it is simply risk as variance of rewards from subjective value (calculated from reward and choice).

Thank you for pointing these issues out. We have fully addressed the points by (i) using different names for the two value terms, (ii) clearly stating the purpose and use of each term, (iii) extending our analysis to test the suggested alternative risk definition – the results are shown in a table and indicate that numbers of risk neurons were very similar for the extended risk definition.

In detail, to rectify the first point, we revised the Results section to clarify for what purposes these definitions were used and, in the Materials and methods, we now use distinct terms for these value definitions. We also clarify that Equation 7 was our main value measure used for behavioral and neuronal analyses whereas Equation 6 was used for comparisons to risk in Figure 3C. These changes to the text are shown below.

With respect to the second point, our neural subjective risk analyses were performed with the risk measure in Equation 8, based on the sum of the weighted, squared deviations from the mean of the object-specific reward distribution. We prefer this risk definition because it is simple (no additional assumptions about how choice history might be incorporated and weighted), directly interpretable (as deviation of reward from the mean of the reward distribution), and follows previous neuronal studies (which tested risk as variance of a reward distribution). We note that it is partly a *conceptual* question of whether choice history should be incorporated into neuronal measures of value or risk, or considered as a separate behavioral influence (e.g. Lau and Glimcher, 2008, modelled choice history for behavior but based their neuronal value measure on reward history without choice history).

Nevertheless, we appreciate that other, more elaborate risk definitions are possible and of interest and therefore include the following new analyses. We added a table (Figure 4—figure supplement 4) to compare the numbers of risk neurons obtained with different risk definitions, including the one suggested by the reviewer (incorporating reward and choice history). These alternative definitions yielded identical or only slightly higher numbers of risk neurons compared to our main risk definition (< 5% variation in identified neurons). We therefore focus on our main risk definition (Equation 8), which is simpler and conservative as it makes fewer assumptions.

Revised Results text: “We followed previous studies of matching behavior (Lau and Glimcher, 2005) that distinguished two influences on value: the history of recent rewards and the history of recent choices. […] Thus, we estimated object value based on both subjectively weighted reward history and subjectively weighted choice history (Equation 7); this constituted our main value measure for behavioral and neuronal analyses.”

Revised Materials and methods text: **“**We followed previous studies of matching behavior (Lau and Glimcher, 2005) that distinguished two influences on value: the history of recent rewards and the history of recent choices. The first object-value component related to reward history,OVAr, can be estimated by the mean of subjectively weighted reward history over the past ten trials (Equation 6):…”

“In tasks used to study matching behavior, such as the present one, it has been shown that choice history has an additional influence on behavior and that this influence can be estimated using logistic regression (Lau and Glimcher, 2005). To account for this second object-value component related to choice history, we estimated a subjective measure of object value that incorporated both a dependence on weighted reward history and a dependence on weighted choice history (Equation 7):…”

New Results text: “We also considered alternative, more complex definitions of subjective risk that incorporated either weighted reward history or both weighted reward and choice history in the risk calculation. […] We therefore focused on our main risk definition (Equation 8), which was simpler and more conservative as it incorporated fewer assumptions.”

New Materials and methods text: “Alternative, more complex definitions of subjective risk in our task could incorporate the weighting of past trials in the calculation of the mean reward (the subtrahend in the numerator of Equation 8) or incorporate both weighted reward history and weighted choice history in this calculation. We explore these possibilities in a supplementary analysis (Figure 4—figure supplement 4).”

The description in the Materials and methods (subsection “Final calculation of subjective risk”) is incorrect given the revised Equation 8, as it implies that β weights are applied to rewards rather than to the squared deviations of rewards from OV_A_ (as described by the revised equation). For example, "βjrRA" term is not in the equation, and "the summed, squared deviation of subjectively weighted reward amounts from the mean weighted value of past rewards)" does not match Equation 8.

Thank you for pointing this out. We have revised the section as follows:

Revised Materials and methods text: “We used the following definition as our main measure of subjective object risk (Equation 8):

varA=∑j=1Nβjr(RAi-j-(∑j=1NRAi-j)/N)2N-1

with βj rrepresenting the weighting coefficients for past rewards (derived from Equation 5), RA as reward delivery after choice of object A, *j* as index for past trials relative to the current *i*th trial, and *N* as the number of past trials included in the model (*N* = 10); the term (∑j=1NRAi-j)/N represents the mean reward over the last ten trials. Thus, the equation derives subjective object risk from the summed, subjectively weighted, squared deviation of reward amounts in the last ten trials from the mean reward over the last ten trials.”